# Structural basis for the ligand promiscuity of the neofunctionalized, carotenoid-binding fasciclin domain protein AstaP

Fedor D. Kornilov[1,2,6], Yury B. Slonimskiy [3,6], Daria A. Lunegova[3,6], Nikita A. Egorkin[3], Anna G. Savitskaya[1], Sergey Yu. Kleymenov[4], Eugene G. Maksimov [5], Sergey A. Goncharuk [1,2], Konstantin S. Mineev [1,2✉] & Nikolai N. Sluchanko [3✉]

Fasciclins (FAS1) are ancient adhesion protein domains with no common small ligand binding reported. A unique microalgal FAS1-containing astaxanthin (AXT)-binding protein (AstaP) binds a broad repertoire of carotenoids by a largely unknown mechanism. Here, we explain the ligand promiscuity of AstaP-orange1 (AstaPo1) by determining its NMR structure in complex with AXT and validating this structure by SAXS, calorimetry, optical spectroscopy and mutagenesis. α1-α2 helices of the AstaPo1 FAS1 domain embrace the carotenoid polyene like a jaw, forming a hydrophobic tunnel, too short to cap the AXT β-ionone rings and dictate specificity. AXT-contacting AstaPo1 residues exhibit different conservation in AstaPs with the tentative carotenoid-binding function and in FAS1 proteins generally, which supports the idea of AstaP neofunctionalization within green algae. Intriguingly, a cyanobacterial homolog with a similar domain structure cannot bind carotenoids under identical conditions. These structure-activity relationships provide the first step towards the sequence-based prediction of the carotenoid-binding FAS1 members.

[1] Shemyakin-Ovchinnikov Institute of Bioorganic Chemistry of the Russian Academy of Sciences, 117997 Moscow, Russia. [2] Moscow Institute of Physics and Technology, 141701 Dolgoprudny, Russia. [3] A.N. Bach Institute of Biochemistry, Federal Research Center of Biotechnology of the Russian Academy of Sciences, 119071 Moscow, Russia. [4] Koltzov Institute of Developmental Biology of the Russian Academy of Sciences, 26 Vavilov Street, 119334 Moscow, Russia. [5] M.V. Lomonosov Moscow State University, Faculty of Biology, 119991 Moscow, Russia. [6]These authors contributed equally: Fedor D. Kornilov, Yury B. Slonimskiy, Daria A. Lunegova. ✉email: konstantin.mineev@gmail.com; nikolai.sluchanko@mail.ru

The fasciclin (FAS1) family comprises many proteins containing the FAS1-like domain (InterPro accession number IPR000782), which is heavily used in extracellular proteins serving as cell adhesion molecules (CAMs). The fasciclin I was first identified in insect CAMs within axon fascicles (hence the name fasciclin)[1] and can mediate homophilic or heterophilic cell adhesion, depending on whether FAS1 interacts with other FAS1 domains or with different CAMs such as integrins on the surface of other cells or with components of the extracellular matrix[2].

FAS1 domain is thought to have appeared in evolution very early and is ubiquitously found in bacteria and yeast to plants and animals[2]. Such domains are typically 140 residues long and have a rather conserved α/β-mixed architecture belonging to a "β-grasp fold" superfamily[2]. They are common in many secreted and transmembrane proteins and can be duplicated in one polypeptide. For example, four copies are found in insect fasciclin I[3], human periostin, and transforming growth factor-β induced protein[2], and seven in human stabilin-1 and stabilin-2[4]. Some proteins contain only a single FAS1 domain (for example, a mycobacterial immunogenic secreted protein MPB70[5], a *Rhodobacter spheroides* protein Fdp[6] or a CupS protein from *Thermosynechococcus elongatus*[7]). Prokaryotes tend to have single-domain FAS1 proteins, whereas eukaryotes typically have such domains organized in tandems. Many FAS1-containing proteins are attached to the membrane and are glycosylated[2].

The first structure of a FAS1 domain was obtained for *Drosophila melanogaster* FAS1[3]. The fold of different FAS1 domains is common despite a rather low sequence identity among the members. All members of the FAS1 superfamily are characterized by the presence of conserved separate 10-15-residue motifs, H1 and H2, as well as a YH dyad in between[2]. Although the predominant number of FAS1 structures was determined for isolated domains, there are seldom structures of multidomain FAS1 proteins such as that of the four-domain human transforming growth factor-β induced protein, mutations in which lead to abnormal self-association and accumulation of deposits in the cornea leading to dystrophy and blindness[8], or periostin, an extracellular matrix protein secreted by fibroblasts and up-regulated in a range of cancers[9]. Nevertheless, the detailed molecular mechanism of homophilic protein–protein recognition in cell adhesion involving FAS1 domains remains elusive.

That FAS1 domains are common in animals and plants became clear when the two FAS1 domain CAMs homologous to *Drosophila* fasciclin I were identified in multicellular alga *Volvox carteri*[10]. Higher plants were also shown to contain two FAS1 domain CAMs[11]. Interestingly, the cell adhesion function of FAS1 domains is so conserved that replacement of an original FAS1 domain by its distant homologs from either single-domain or multidomain FAS1 proteins does not abolish cell adhesion[12]. At the same time, no common ligand-binding function of the FAS1 domains has been reported.

In 2013, a remarkable protein was identified in green algae[13]. While it was found as a stress-inducible protein containing a predicted FAS1 domain, it was isolated from the natural source as a soluble astaxanthin (AXT)-binding protein (hence called AstaP) undergoing glycosylation[13]. While the preprotein contained a predicted signal peptide for secretion, it was accumulated on the cell periphery in response to hyperinsolation and osmotic stress (often associated with drought conditions), to perform a photoprotective role, presumably, via direct quenching of reactive oxygen species (ROS) and also via high-energy light filtering effects[13]. Unlike the cyanobacterial photosensory Orange Carotenoid Protein (OCP)[14], AstaP did not undergo a photo-induced transformation even after intense illumination[15]. Several orthologs of AstaP have been isolated from *Scenedesmus* sp. Oki-4N and *Scenedesmus costatus* SAG 46.88 and identified in other *Scenedesmaceae* species; those

orthologs had either one or two FAS1 domains, differed by the pI and the presence of glycosylation sites[16,17]. Based on differences in the absorbance spectrum, AstaP orthologs were classified into orange or pink subgroups (for example, *Scenedesmus* sp. Oki-4N expresses one orange, AstaP-orange2, and two pink AstaP proteins, AstaP-pink1 and AstaP-pink2)[16]. The recently described rather broad distribution of the new FAS1 family member, AstaP, in many *Scenedesmaceae* species provided compelling evidence for its biologically relevant carotenoid-binding function[17]. Yet, the carotenoid specificity had remained unaddressed.

AstaP was isolated from algal cells mainly as an AXT-bound form[18]. However, focusing on the first AstaP described, from *Coelastrella astaxanthina* sp. Ki-4 (AstaP-orange1, or AstaPo1 for short)[13,16], we have recently demonstrated that the recombinant AstaPo1 can bind a uniquely broad repertoire of carotenoids, which includes not only xanthophylls such as canthaxanthin (CAN) and zeaxanthin (ZEA), but also non-oxygenated β-carotene (βCar)[15]. AstaPo1 holoforms could be efficiently reconstituted in vitro by mixing with pure carotenoids or by expression in special carotenoid-synthesizing *E. coli* cells and could deliver the bound carotenoid to the apoforms of unrelated carotenoproteins or to biological membrane models[15]. While these results expanded the toolkit of known carotenoproteins for carotenoid solubilization and targeted delivery, the structural basis for the carotenoid binding mechanism by AstaP, and potentially other FAS1 proteins, remained enigmatic. The apparent lack of carotenoid specificity also remained puzzling.

To fill these gaps, here we study the AstaPo1 structure in complex with carotenoid and structure-function relationships, which suggest the carotenoid capture mechanism and provide the first step toward the sequence-based prediction of the neofunctionalized carotenoid-binding ability in AstaP homologs.

## Results

**Localization of the ligand-binding site within AstaPo1.** AstaPo1 has been shown to bind different xanthophyll types and β-carotene[15] (Fig. 1a). We first wanted to localize the carotenoid-binding region within the mature AstaPo1 (residues 21–223, no signal peptide). Considering the presumable domain structure of this protein (Fig. 1b) and assuming almost equivalent binding of xanthophylls AXT, CAN, and ZEA[15], we prepared AstaPo1 mutants lacking either the N-terminal region, or the C-terminal region, or both, and selected to produce them in ZEA-synthesizing *E. coli* cells, which were known to yield stable ZEA-bound non-truncated AstaPo1[15]. Analytical size-exclusion chromatography (SEC) with continuous absorbance spectrum detection confirmed that truncations lead to a stepwise reduction of the protein monomer size (Fig. 1c), but do not affect the absorbance spectrum in the visible region. It retained the intact vibronic structure of the ZEA-bound AstaPo1 (Fig. 1d) even in the case of the shortest variant corresponding to the central domain, residues 41–190. A high Vis/UV absorbance ratio indicated unaltered and highly efficient carotenoid binding. Both 21–190 and 41–190 variants of AstaPo1 had the most pronounced Vis/UV ratio (~4, equal for both variants), which apparently reflected the removal of C-terminal Trp residues, W197 and W201 (Fig. 1b).

Figure 1e shows that ZEA binding affected the intrinsic Trp fluorescence of AstaPo1. Having three Trp residues (W79, W197, and W201), the AstaPo1 apoform exhibited a high-intensity Trp fluorescence, which was significantly quenched in the ZEA-bound form. Interestingly, the fluorescence maximum remained unchanged at 347 nm, which corresponds to partially exposed Trp residues (Fig. 1e)[19]. Steady-state Trp fluorescence of the shortest ZEA-bound AstaPo1 variant, residues 41–190, which corresponds to the tentative FAS1-like domain containing a single

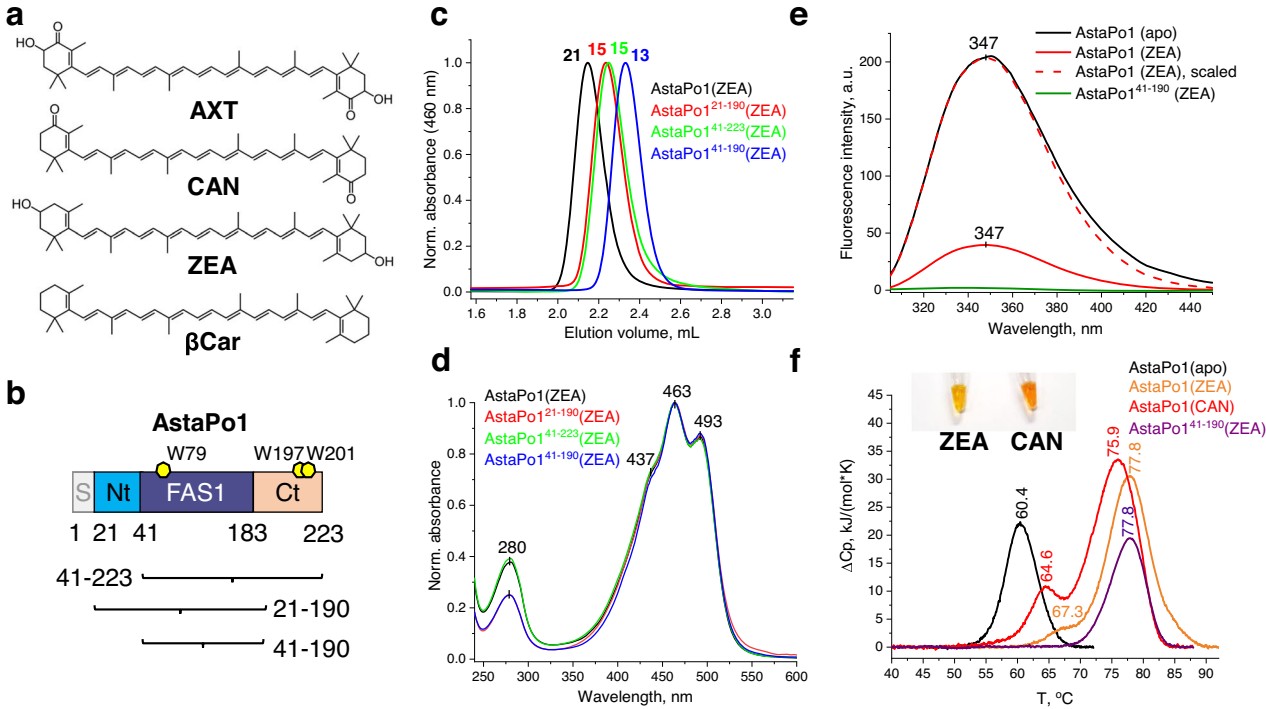

**Fig. 1 Localization of the carotenoid-binding region of AstaPo1. a** Structural formulae of several carotenoids which can be bound by AstaPo1. AXT astaxanthin, CAN canthaxanthin, ZEA zeaxanthin, βCar β-carotene. **b** The domain structure of AstaPo1 showing the location of the predicted FAS1-like domain and the flanking N- and C-terminal regions. Note that the AstaPo1 preprotein contains an N-terminal signal peptide (residues 1–20), which was omitted in our construct as this peptide hampers the expression and purification of the soluble protein. Instead, the N terminus of the protein carried artificial residues [17]GPHM[20] left after the His$_6$-tag removal. **c, d** Analytical SEC profiles with diode-array detection, revealing the effect of AstaPo1 truncations on the size of the protein monomer (**c**) and its absorbance spectrum (**d**). The apparent $M_w$ values determined from column calibration are indicated in kDa. **e** Intrinsic Trp fluorescence spectra of AstaPo1 showing the effect of ZEA binding to the wild-type and truncated AstaPo1. The excitation wavelength was 297 nm. **f** DSC thermograms showing the effect of carotenoid binding and truncation of the N- and C-terminal regions on the thermal stability of AstaPo1. $T_m$ values are indicated as determined from the maxima of the peaks. The heating rate was 1 °C per min. The insert shows the color of the AstaPo1 samples with ZEA or CAN used.

Trp79 residue, was completely quenched likely by a non-radiative energy transfer mechanism to the carotenoid (Fig. 1e), which confirms the direct carotenoid interaction within this domain. Given that Trp residues of carotenoproteins often form direct contact with carotenoids[20–22], one may expect a similar involvement of Trp79.

Using differential scanning calorimetry (DSC), we found that the AstaPo1 apoform had a rather high denaturation temperature ($T_m$ ~ 60 °C), whereas carotenoid binding substantially enhanced it further (to $T_m$ of 75–80 °C) (Fig. 1f). Most importantly, truncation of the N- and C-terminal domains did not lower the high $T_m$ for the AstaPo1 holoform (Fig. 1f).

Collectively, our data implied that the carotenoid-binding region of AstaPo1 is contained within its predicted FAS1 domain. Of note, while the apoform of the shortest AstaPo1 variant (41–190) corresponding to the FAS1 domain was entirely found in inclusion bodies, its holoform could be obtained upon expression in ZEA-producing *E. coli* cells. This suggests that the N- and C-terminal regions of AstaPo1 and carotenoid binding dramatically improve protein solubility. Successful formation of the AstaPo1$^{41–190}$ holoform in *E. coli* cells indicates that such miniaturized AstaPo1 variant retains the ability to extract carotenoids from membranes in the absence of the N- and C-terminal regions, which are therefore not essential for this function. Given the tiny size of this water-soluble protein (16 kDa), it may be promising for various biomedical applications.

Nevertheless, the contributions from the tail regions to carotenoid embedment by AstaPo1 remained to be clarified by structural investigation in the context of the non-truncated protein.

**Solution structure of AstaPo1 in complex with carotenoid.** To investigate the structure of the AstaPo1 complex with its major native carotenoid, AXT, we took advantage of solution nuclear magnetic resonance (NMR) spectroscopy. We expressed the $^{13}$C/$^{15}$N-labeled AstaPo1 apoprotein and complexed it with the unlabeled chemically pure AXT. After extensive screening, we found the conditions allowing us to obtain AstaPo1(AXT) samples with 280 μM protein concentration (containing an equimolar concentration of AXT) and acquire the high-quality NMR spectra (Supplementary Fig. 1).

NMR chemical shifts of AXT inside the protein were elucidated using the isotope-filtered Nuclear Overhauser Effect spectroscopy (NOESY) experiments[23]. Such analysis revealed the presence of four sets of NMR signals, compared to the spectra of AXT acquired in organic solvents (Supplementary Fig. 2). One additional set of signals may arise due to the protein-induced asymmetry of the local environment around the two initially symmetrical parts of the AXT molecule (Fig. 2a). The other signals are most likely the consequence of conformational heterogeneity, which is in agreement with peak splittings observed in the protein NMR spectra (Supplementary Fig. 3). We analyzed several options that could explain this heterogeneity, including the alternative orientation of AXT within the cavity, cis-trans isomerism of double bonds[24], or the R-S isomerism of

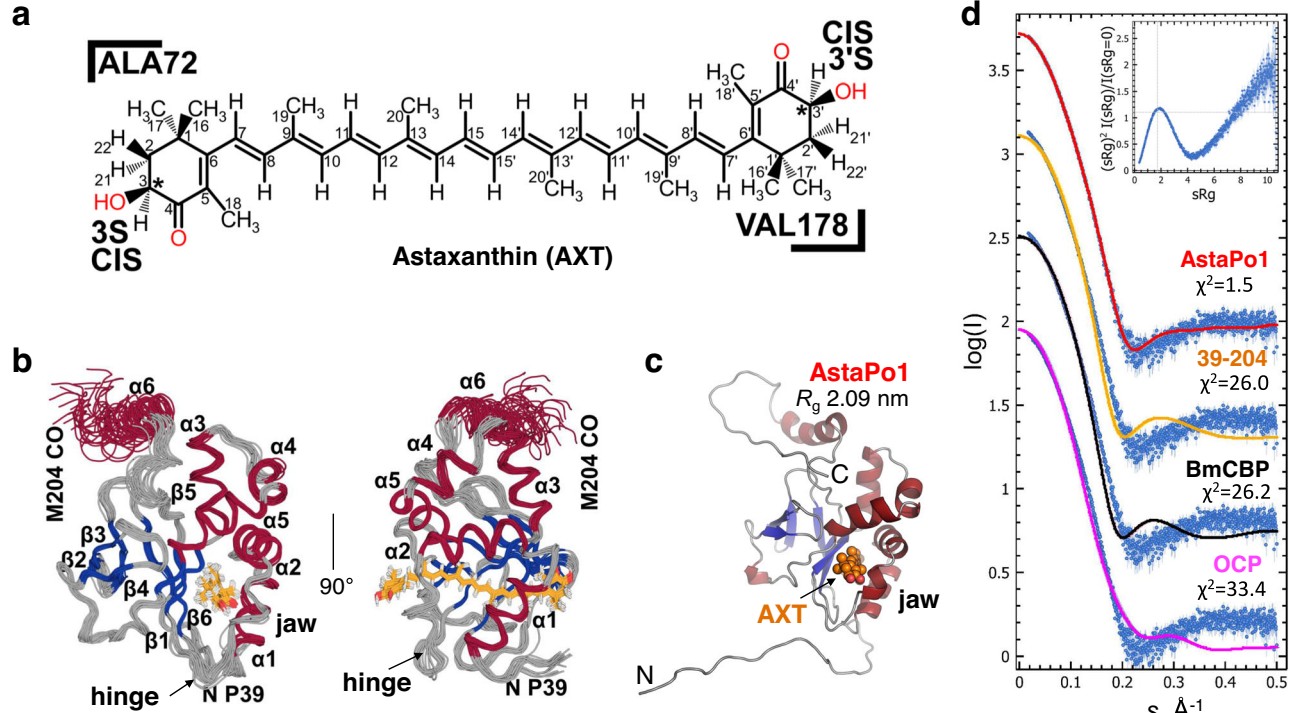

**Fig. 2 Spatial structure of the AstaPo1(AXT) complex. a** Chemical structure of the AXT stereoisomer used for NMR structure determination, with the indication of atom numbering. Asterisks indicate the chiral centers. Oxygenated groups are in red. **b** Twenty NMR structures of AstaPo1(AXT) complex with the lowest restraint violations, shown superimposed over the backbone atoms of secondary structure elements (except for the C-terminal helix α6, residues 191–204) in two orthogonal views. The unstructured tails are hidden for clarity. Elements of the secondary structure are labeled. The AXT molecule is shown by orange sticks. **c** A single representative full-atom NMR structure of AstaPo1(AXT) provides an excellent fit to the SAXS data collected for the AstaPo1(ZEA) complex ($R_g$ of the model is 2.09 nm, $\chi^2 = 1.5$; at 7 mg mL$^{-1}$). $R_g$ of the twenty NMR full-atom structural models ranged from 1.90–2.56 nm. Secondary structures are color-coded (red—α-helices, blue—β-strands, and gray—unstructured regions). **d** The fits to the SAXS data from the best-fitting NMR structure of the full-atom AstaPo1(AXT) complex, from the AstaPo1 FAS1 domain (residues 39–204; model $R_g$ 1.85 nm), from the *Bombyx mori* Carotenoid-Binding Protein (model $R_g$ 1.85 nm)[22], and from the *Synechocystis* Orange Carotenoid Protein (model $R_g$ 2.05 nm)[68]. The quality of the fits is indicated as $\chi^2$ for each case. The insert shows the dimensionless Kratky plot for the AstaPo1(ZEA) complex. Dashed lines show the position of the maximum for the rigid sphere, which is given for reference.

AXT, which is observed with respect to the position 3 (3′) of the β-ionone rings[25], and revealed that the latter phenomenon does actually take place (Supplementary Text 1 and Supplementary Figs. 4–6). Therefore, we studied the spatial structure of the AstaPo1(AXT) complex only for one of the states, (3S and 3′S)-AXT, since the basic structural data for this isomer may be extracted from the PDB[26,27].

Following the manual analysis of all the intermolecular NMR contacts, we found 150 intermolecular distance restraints and resolved the spatial structure of the AstaPo1(AXT) complex (Fig. 2b and Table 1). The central part of the protein adopts a typical fold of a FAS1 domain[3,5] with a wedge-shaped β-sandwich, consisting of two three-stranded β-sheets and six α-helices. While five helices are typical for the FAS1 fold, the sixth (residues D191-E202), rather solvent-exposed C-terminal helix, is peculiar in not being well packed into the globule with the other secondary structure elements, although it is anchored by the π-cation and ionic interactions between the W197 and E200 residues and charged side chains of α3 helix (residues N86–L96). Using the NMR relaxation parameters, we analyzed the dynamics of the AstaPo1 backbone and revealed that helix α6 tumbles as a whole with the protein, with some local mobility observed only at its C-terminal part (Supplementary Figs. 7, 8). In contrast, we observed high-amplitude fast motions for the N- and C-terminal regions (residues 17–32 and 204–223) flanking the FAS1 domain. The structure of these parts of AstaPo1 is not well-defined in the NMR set, suggesting that the terminal residues are disordered,

which is in line with the idea that they are chiefly required to maintain protein solubility.

To justify the NMR structure, we exploited small-angle X-ray scattering (SAXS) of AstaPo1 complexes with either ZEA or CAN, as these particular holoproteins could be most efficiently produced in *E. coli* expressing the corresponding carotenoids. Despite the two distinct carotenoid types, differences in protein concentration, and variances in the residual amount of the apoform present in the samples (Fig. 1f), the SAXS data for AstaPo1(ZEA) and AstaPo1(CAN) yielded very similar structural parameters in solution (Table 2). Therefore, the type of bound carotenoid does not affect the protein conformation, and the SAXS data for the least noisy AstaPo1(ZEA) SAXS curve is representative also for AstaPo1(AXT). One of the NMR models provided an excellent fit ($\chi^2 = 1.5$) to the SAXS data without any additional assumptions, whereas structural models of the individual FAS1 domain or of the other carotenoproteins of similar size ($R_g \sim 2$ nm) provided inadequate fits (Fig. 2c, d). Of note, the dimensionless Kratky plot for AstaPo1(ZEA) revealed the characteristic bell-shape and a gradual rise of the curve at large $sR_g$ values (Fig. 2d, insert), which reflected a nearly spherical folded protein with a number of peripheral flexible regions. Such a description perfectly agrees with our NMR data.

**Carotenoid-binding mode and protein–pigment interactions.** NMR data revealed that AXT binds to AstaPo1 at the β-sheet

**Table 1 NMR input data and statistics for the AstaPo1(AXT) complex.**

| NMR distance and dihedral constraints | |
|---|---|
| Distance constraints | |
| Total NOE for AstaPo1 | 1671 |
| Intra-residue | 447 |
| Inter-residue | 1224 |
| Sequential ($\|i-j\| = 1$) | 481 |
| Medium-range ($\|i-j\| \leq 4$) | 283 |
| Long-range ($\|i - j\| > 5$) | 460 |
| Intermolecular NOE | 150 |
| Hydrogen bonds | 52 |
| Total dihedral angle restraints | |
| $\varphi$ | 151 |
| $\psi$ | 155 |
| $\chi_1$ | 41 |
| Constraints per residue (excluding unstructured parts) | 15.1 |
| Structure statistics (for a set of 20 best structures) | |
| CYANA target function | $1.62 \pm 0.16$ |
| Violations (mean ± SD) | |
| Distance mean [max], (Å) | $0.0056 \pm 0.0009$ [$0.23 \pm 0.07$] |
| Dihedral angles mean [max], (°) | $0.382 \pm 0.042$ [$2.62 \pm 0.74$] |
| RMSD of atom coordinates (stable secondary structure, 80 residues) (mean ± SD) | |
| Backbone (Å) | $0.57 \pm 0.04$ |
| Heavy (Å) | $1.32 \pm 0.10$ |
| RMSD of atom coordinates (structured core, residues 40–203) (mean ± SD) | |
| Backbone (Å) | $1.45 \pm 0.39$ |
| Heavy (Å) | $1.92 \pm 0.30$ |
| RMSD of atom coordinates, AXT, (Å) | $0.31 \pm 0.08$ |
| Ramachandran plot outliers | |
| Residues in most favored regions | 94% |
| Residues in additionally allowed regions | 6% |
| Residues in generously allowed regions | 0% |
| Residues in disallowed regions | 0% |
| PDB ID | 8C18 |

**Table 2 SAXS-derived structural parameters of the two AstaPo1 holoforms.**

| Parameter | AstaPo1(ZEA) | AstaPo1(CAN) |
|---|---|---|
| $M_w$ calculated from the sequence | 21.6 kDa (apo) | 21.6 kDa (apo) |
| Number of residues | 207 | 207 |
| Protein concentration | 7.05 mg mL$^{-1}$ | 2.57 mg mL$^{-1}$ |
| $R_g$ (Guinier) | 2.09 ± 0.01 nm* | 2.13 ± 0.01 nm |
| s$R_g$ limits | 0.39–1.29 | 0.32–1.29 |
| $R_g$ (reverse) | 2.11 ± 0.03 nm | 2.18 ± 0.04 nm |
| $D_{max}$ | 7.8 nm | 8.1 nm |
| Porod volume | 38.1 nm$^3$ | 40.8 nm$^3$ |
| $M_w$ Porod | 23.8 kDa | 25.5 kDa |
| $M_w$ MoW | 14.3 kDa | 15.5 kDa |
| $M_w$ Vc | 21.1 kDa | 22.2 kDa |
| $M_w$ (size and shape) | 23.2 kDa | 25.3 kDa |
| Kratky plot | bell-shaped (folded) | bell-shaped (folded) |
| SASBDB ID | SASDRG8 | SASDRH8 |

*All SAXS-derived parameters were calculated using the ATSAS 2.8 software package[54].

composed of β1-β6-β5 strands and is covered by a jaw-like structure formed by two N-terminal helices α1, α2, and the α2-β1 hinge loop (Figs. 2b, 3a). In the observed binding mode, the hydrophobic carotenoid polyene is packed against the apolar side chains decorating the tunnel (at least 14 Ile/Val/Leu residues) (Fig. 3a–e), many of which are well conserved and belong to the FAS1-specific H1, H2, and YH motifs (Fig. 3f). At the same time, the β-ionone rings of the carotenoid protrude from the protein globule and are solvent-exposed, which explains the lack of fine structure in the absorption spectra of AstaPo1-bound AXT or CAN (Fig. 3b, e)[15]. Surprisingly, we did not observe any specific protein-pigment interactions, such as hydrogen bonds or π-stacking found in other carotenoproteins with the known structure[20–22,26]. The only transient polar contact may take place between the side chain of Q56 and hydroxyl or carbonyl groups of AXT (Fig. 3c, d). Obviously, this contact cannot be formed in the case of β-carotene, one of the established AstaPo1 ligands[15]. Hence, Q56 is perhaps not a strong carotenoid-binding determinant, although it could explain the apparently more efficient AstaPo1 binding with xanthophylls[15]. While this residue is highly variable in the FAS1 domain-containing proteins (Fig. 3f), its chemical functionality is clearly preferred in AstaP orthologs with the proposed carotenoid-binding capacity[13,16–18] (see also below).

We did not find any special contribution from the disordered tail regions into the interface with AXT, which supports the conclusions made above for the truncated AstaPo1. Since the main protein-pigment interactions revealed by the AstaPo1(AXT) structure are the hydrophobic contacts holding the polyene chain of the carotenoid in the conserved hydrophobic tunnel of the AstaPo1 structure, we identify no features that would determine the ligand specificity. Thus, the protein should be able to bind any sufficiently long rod-like hydrophobic molecules with the optional presence of polar headgroups.

According to the low-amplitude Vis/UV circular dichroism (CD) spectra, the carotenoid molecule in AstaPo1 is rather symmetric and straight, in contrast to the curved conformation previously reported for BmCBP[22] or OCP[28] carotenoproteins (Fig. 3g). This supports the notion that the AstaPo1-bound AXT does not experience any significant bending or torsion, which is often a consequence of direct chemical contacts with the carotenoid rings, such as in OCP[20] or in β-crustacyanin[26]. In nice agreement with the Trp fluorescence quenching data (Fig. 1e), our NMR structure reveals that one of the AXT rings neighbors the side chain of the single Trp79 in the carotenoid-binding domain of AstaPo1, although its conformation and distance (>5 Å) from the ring are incompatible with H-bonds or π-stacking interactions that would stabilize the bound carotenoid (Fig. 3a). Interestingly, Trp in this position is not fully conserved in the FAS1 protein superfamily, and is often replaced by a Phe (Fig. 3f). Crucially, the amino acid substitution W79F in AstaPo1 produced in ZEA-synthesizing E. coli cells, does not abolish its ZEA-binding capacity nor change the absorbance spectrum (Fig. 3h), which supports the idea that the indolyl group in this position is dispensable for carotenoid binding. To test the possible importance of the tunnel residues, we also obtained the I176F mutant with a Phe residue introduced in the conserved H2 motif in the middle of the tunnel (Fig. 3f) and produced it in ZEA-synthesizing E. coli cells. While it surprisingly retained the carotenoid-binding ability, its efficiency was compromised (only 60% of the holoform), as judged from a substantially lowered Vis/UV absorbance ratio of 1.7 instead of 2.8 for the wild-type (Fig. 3h). Curiously, the Vis absorbance of ZEA bound to the I176F mutant was appreciably blue-shifted by ~4–7 nm, with an altered fine structure, compared with the wild-type AstaPo1 (Fig. 3h). Such spectral change reflects the renowned sensitivity of the carotenoid absorbance to even subtle changes in microenvironment[22] and the

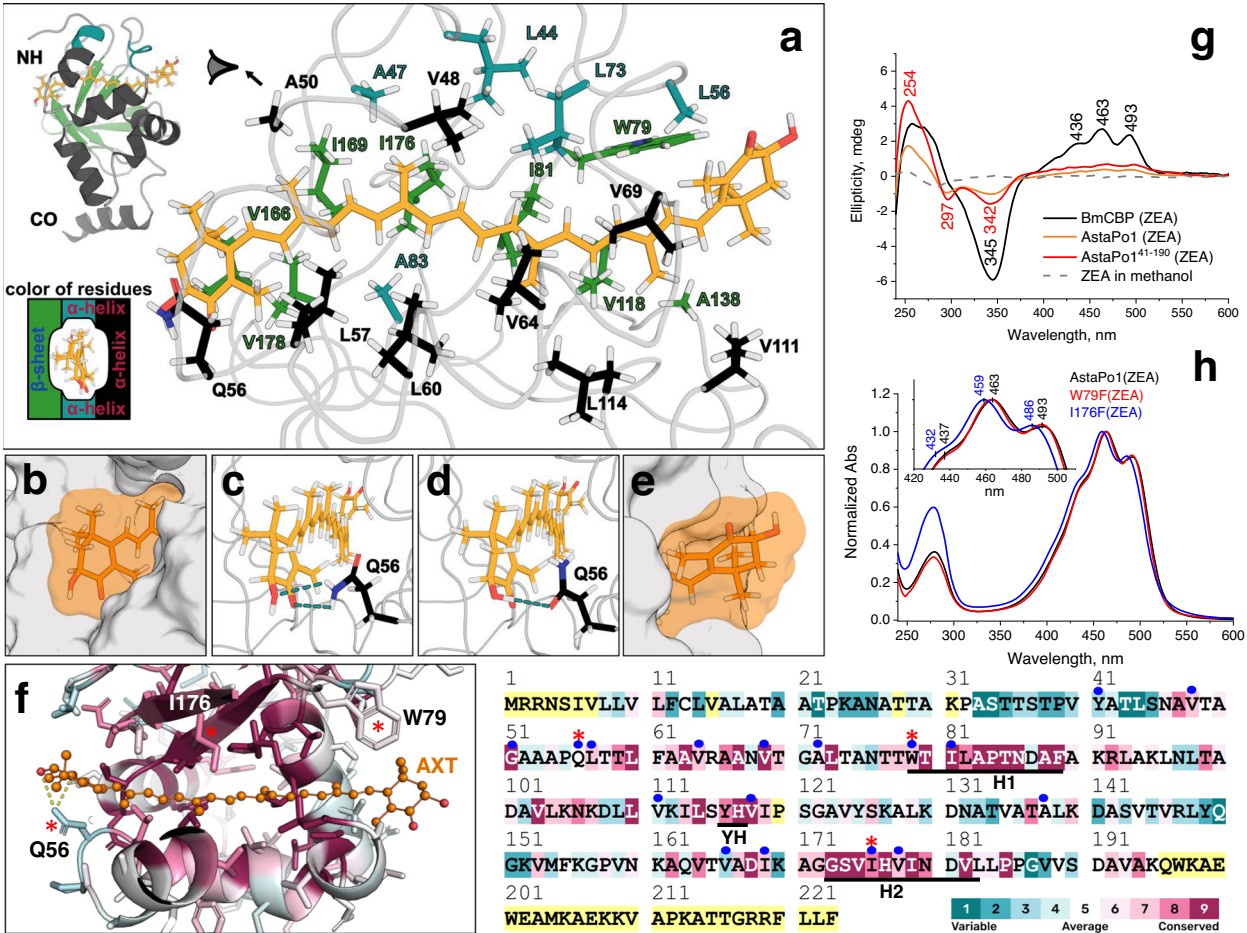

**Fig. 3 AXT binding mode and AstaPo1–pigment interactions. a–e** Different views of the carotenoid binding tunnel of AstaPo1. The coloring scheme in panel **a** is indicated. **b, e** Polar headgroups of AXT exiting the protein molecule from both sides of the carotenoid-binding tunnel. The Connolly surface with solvent radius 1.4 Å of AstaPo1 and the carotenoid is shown in gray and orange, respectively. **c, d** Polar contacts formed between the AXT polar groups and the side chain of the non-conserved Q56 residue of AstaPo1. **f** Amino acid conservation mapped onto the tertiary (left) and primary structure of AstaPo1 (right), according to Consurf[69] analysis of 150 FAS1 domain-containing homologs using the scale shown in the bottom right corner (yellow color indicates positions with insufficient data). The full AstaPo1 sequence is presented for the numbering consistency (residues 1–223). Red asterisks mark the peculiar positions discussed in the text. Blue circles highlight 17 residues whose solvent-accessible surface area changes by at least 10 Å$^2$ upon AXT binding. FAS1-specific conserved motifs H1, H2, and YH are indicated by black bars. **g** Vis-UV CD spectra of AstaPo1(ZEA) and its truncated variant AstaPo1$^{41-190}$(ZEA) as compared with the spectra of BmCBP(ZEA) and free ZEA in methanol. The main extrema are indicated in nm. **h** Absorbance spectra of AstaPo1, its W79F and I176F variants purified from *E. coli* cells synthesizing ZEA.

effect of the bulky Phe side chain in close vicinity of the carotenoid polyene.

**Insights into the carotenoid capture mechanism.** To get insight into the mechanism of the carotenoid uptake by AstaPo1, we undertook the structure investigation of its apo state. The poor quality of NMR spectra and limited stability of the apoprotein did not allow direct structure determination. However, we managed to obtain the partial (63%) assignment of the NMR chemical shifts (Supplementary Fig. 9), which covered the N- and C-terminal tails, helices α3, α4, α5 (partially) and α6, strands β2–β6 (Fig. 4a). According to the secondary chemical shifts, for the covered regions, we observe no difference between the apo and holo states (Supplementary Fig. 10), implying that the structure of β-sheet and helices distant from the ligand-binding site is preserved. On the other hand, the NMR signals of the α1-α2 jaw and adjacent loops are either not seen or substantially broadened, indicating that this AstaPo1 part undergoes conformational exchange, which is rather slow in the NMR chemical shift timescale. In other words, the α1-α2 jaw becomes mobile in

the µs-ms timescale and may detach from the globular core of the protein. These motions may be utilized by the protein to capture the carotenoid molecule from its lipid environment.

A survey of the published FAS1 structures helped to find a peculiar NMR structure of the single-domain FAS1 protein CupS from *T. elongatus* (NAD(P)H dehydrogenase type 1 (NDH-1) complex sensory subunit, Uniprot Q8DMA1: 30.8% sequence identity with AstaPo1 FAS1)[7] (Fig. 4b). When superimposed onto our AstaPo1(AXT) structure using the highly conserved H1/H2 motifs and YH dyads, the CupS structure is remarkably dissimilar by the position of the α1-α2 jaw, suggesting an imaginary rotation over the α2-β1 hinge loop which would resemble a conformational transition from an open-like to the closed, AXT-bound conformation. Of note, such a transition would not involve significant secondary structure rearrangements. We confirmed this using far UV CD spectroscopy of the apo- and carotenoid-bound forms of AstaPo1 (Fig. 4c). Due to the unavoidable contribution of carotenoid to the experimental CD spectrum of the AstaPo1 holoform, the latter was calculated from NMR models of AstaPo1(AXT) using PDBMD2CD[29]. The similarity of

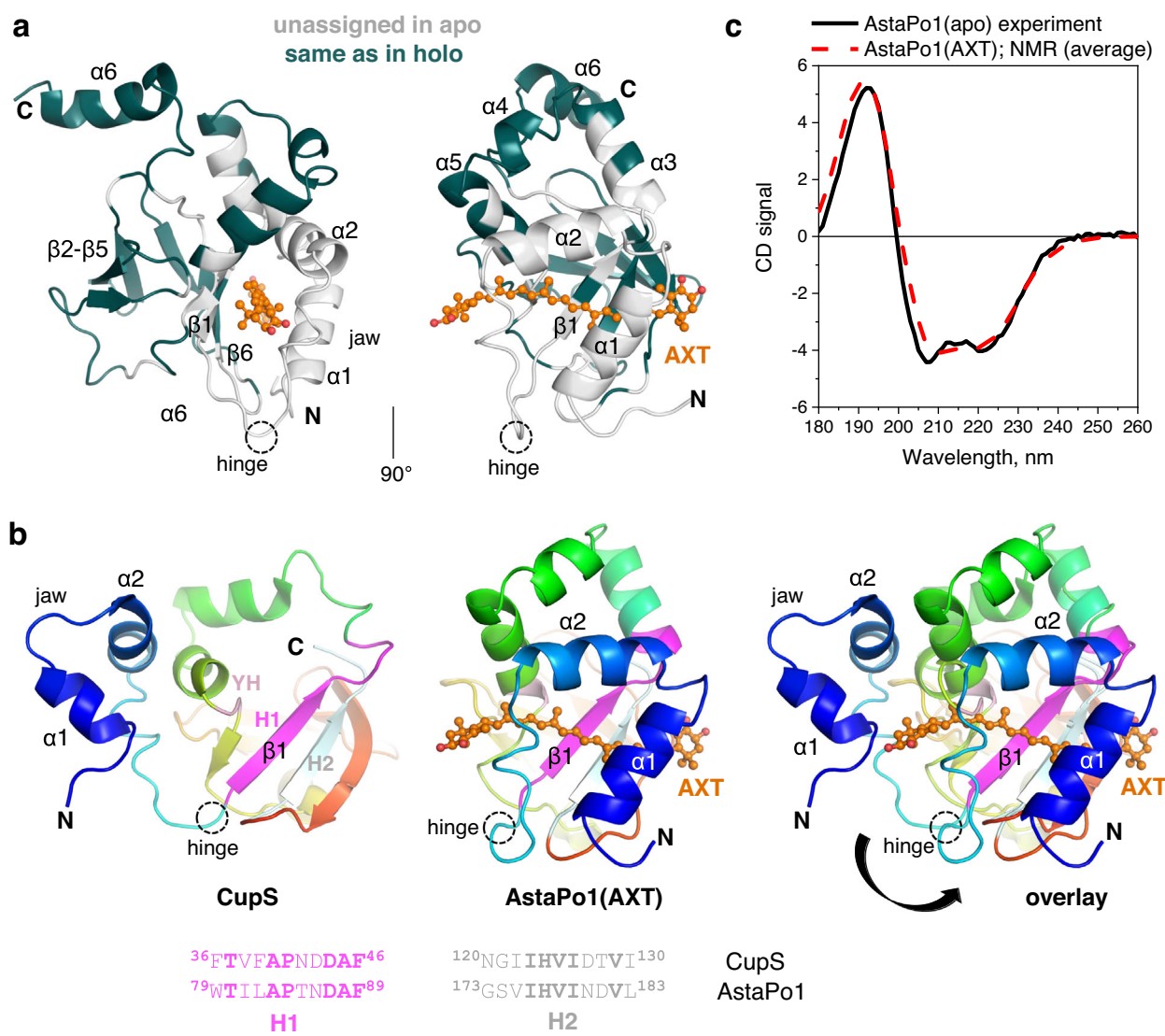

**Fig. 4 Insights into the carotenoid capture mechanism by AstaPo1. a** Structure of the AstaPo1(AXT) complex is colored according to the NMR data obtained for the apoform. Regions with known NMR chemical shifts, which reveal the identical structure with the AstaPo1 holoform, are colored green (Supplementary Fig. 10), and the other parts of the protein, for which no assignment could be obtained, are colored light gray. **b** A hypothetical carotenoid capture mechanism. Three panels represent from left to right: the CupS FAS1-containing protein (PDB ID: 2MXA[7]) in a tentative "open-like" conformation, AstaPo1 in complex with AXT (our NMR structure), and their overlay. The structures are colored by a gradient from blue (N) to red (C), except for the conserved motifs used for structural alignment—H1 (magenta), H2 (pale cyan), and YH (pale pink). The pairwise sequence alignment of the H1 and H2 fragments is shown below (identical residues are in bold font). AXT is shown as an orange ball-and-stick model, the main secondary structure elements and the hinge loop are labeled. The proposed conformational transition from an open to a carotenoid-bound state is depicted by the arrow. **c** Far UV CD spectrum of the AstaPo1 apoform (black line) is shown overlaid with the average CD spectrum calculated from 20 NMR models of the holoform of the same construct (red dashed line).

the apo/holo CD spectra supports the idea of jaw movement as a whole. While remaining speculative, such rearrangement provides a useful insight into the carotenoid capture by AstaPo1.

Thus, we conclude that, while the apoform is not different from the AXT-bound state of AstaPo1 in terms of the overall secondary structure, the flexibility and loop conformations of the α1-α2 jaw and its surroundings change dramatically in response to the ligand uptake.

**Carotenoid binding as a privilege function of a subset of the FAS1 domain-containing proteins.** FAS1 domains have an evolutionarily ancient and widespread fold; however, the ligand-binding capacity of such proteins has not been reported, until recently[13]. Therefore, the possible structural and sequence

determinants of such a new FAS1 function warranted analysis. Based on the NMR structure, we identified AstaPo1 residues directly involved in carotenoid binding by calculating the solvent-accessible surface area (SASA) changes with and without carotenoid (using 10 Å[2] as cutoff) and mapped them onto the primary and secondary structure of AstaPo1 (Fig. 3f and Supplementary Fig. 11). By comparing how these residues are conserved in FAS1 domain-containing proteins in general (Fig. 3f) and in tentative AstaP orthologs in *Scenedesmaceae* with the reported or expected carotenoid-binding capacity[13,16–18] (Fig. 5a), we revealed that many positions are occupied by identical or similar residues in both protein groups, namely, 48, 51, 57, 64, 69, 79, 81, 118, 169, 176, and 178 (AstaPo1 residue numbering). Most of these residues decorate the carotenoid-

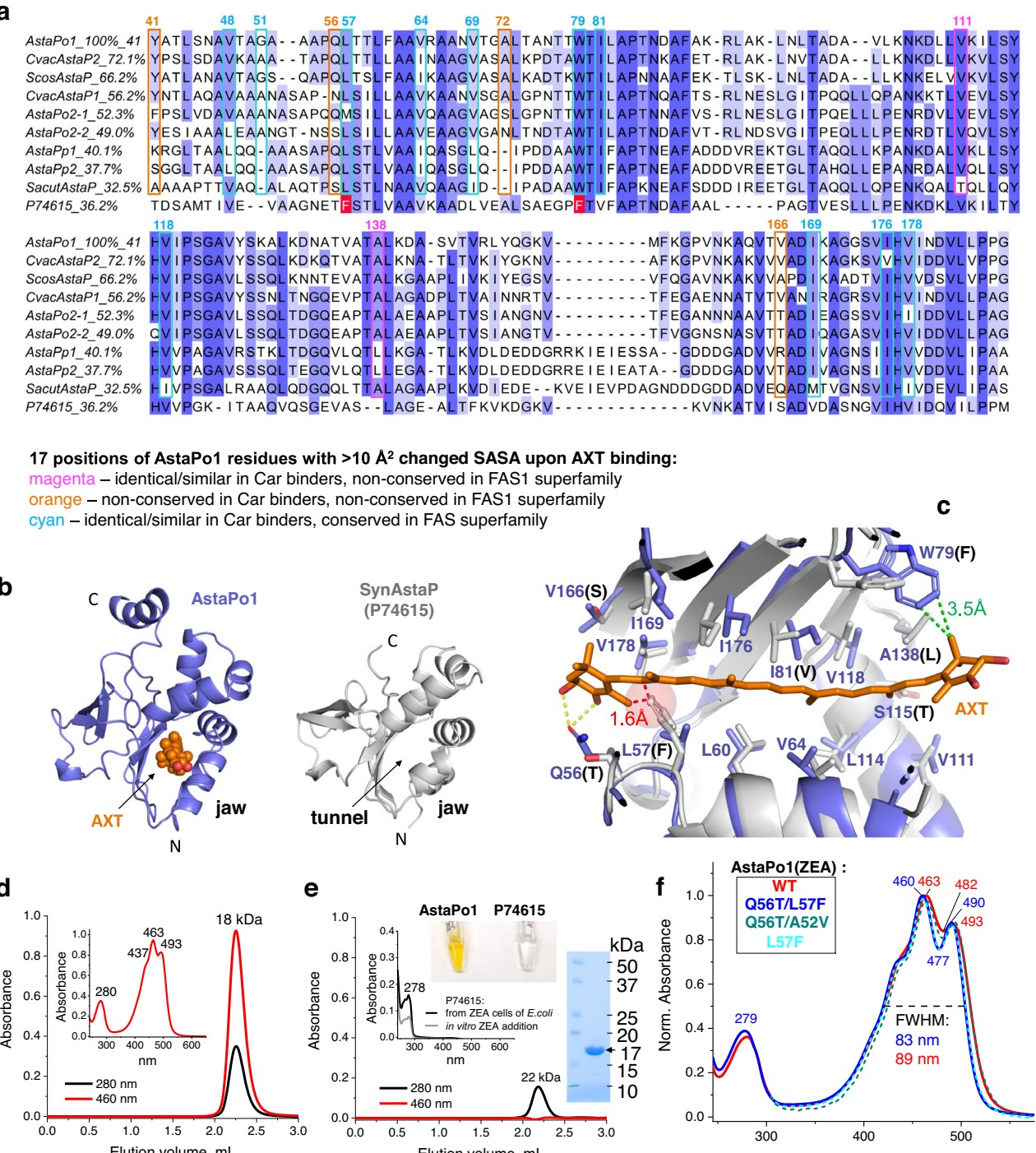

**17 positions of AstaPo1 residues with >10 Å² changed SASA upon AXT binding:**

magenta – identical/similar in Car binders, non-conserved in FAS1 superfamily
orange – non-conserved in Car binders, non-conserved in FAS1 superfamily
cyan – identical/similar in Car binders, conserved in FAS superfamily

binding tunnel of AstaPo1, which is preserved in many FAS1 members regardless of their prospective ligand-binding ability. Positions 41, 56, 72, and 166 are variable in each group, which means that their chemical functionalities are unlikely to be specifically tailored for carotenoid binding. Given their location on the periphery of the hydrophobic tunnel, these residues, even in genuine carotenoid-binding AstaP orthologs, likely play a secondary role in ligand binding, such as in the case of Q56, whose side chain forms transient H-bonds with AXT rings (Fig. 3), but cannot be relevant for binding non-oxygenated β-carotene[15]. Among ten tentative carotenoid-binding AstaP homologs, the glutamine-56 is conserved at least in six, and in one more case is replaced by a synonymous asparagine (Fig. 5a). Whether the

replacement of Gln/Asn by another polar residue in other AstaP homologs inhibits carotenoid binding requires a separate investigation (vide infra).

Most intriguing, some carotenoid-contacting residues in AstaPo1 are identical or similar in AstaP orthologs but are not conserved throughout the FAS1 superfamily (namely, positions 111 and 138 at one of the tunnel exits, see Figs. 5a, 3f). This favors the idea that the carotenoid-binding function is a feature of only a subset of such proteins, but experimental evidence on carotenoid binding by AstaP homologs outside the *Scenedesmaceae* group was missing. To fill this gap, we selected a relatively distant AstaPo1 homolog with an unknown function from the cyanobacterium *Synechocystis* sp. PCC 6803 (UniProt

**Fig. 5 Carotenoid binding as neofunctionalization of AstaP orthologs. a** Multiple sequence alignment (MSA) of the FAS1 domains of the tentative carotenoid (Car) binding AstaP orthologs from *Coelastrella astaxanthina* (AstaPo1; BAN66287.1), *Coelastrella vacuolata* (CvacAstaP1, QYF06643.1; CvacAstaP2; QYF06644.1), *Scenedesmus costatus* (ScosAstaP, QYF06645.1), *Scenedesmus* sp. Oki-4N (AstaPo2-FAS1 domain 1 and AstaPo2-FAS1 domain 2, BBN91622.1; AstaP-pink1, BBN91623.1; AstaP-pink2, BBN91624.1), *Scenedesmus acutus* (SacutAstaP; ACB06751.1), and *Synechocystis* sp. PCC 6803 (SynAstaP, Uniprot P74615), shown with coloring by similarity as shades of blue. NCBI accession numbers are indicated for each sequence (except SynAstaP) in parentheses. The sequence identity of each ortholog relative to AstaPo1 is indicated in %. Seventeen positions of the AstaPo1 sequence whose solvent-accessible surface area (SASA) changed >10 Å$^2$ upon AXT binding (NMR data) are mapped on the MSA to reveal positions satisfying the criteria indicated by magenta, orange, and cyan. Note that positions of Leu57 and Trp79 of AstaPo1 are occupied by Phe residues in SynAstaP (P74615) (marked in red). **b** Comparison of the NMR structure of AstaPo1 complexed with AXT and the Alphafold model of *Synechocystis* ortholog P74615, showing the similar FAS1-like fold featuring the jaw and the tunnel. **c** Structure alignment of AstaPo1(AXT) complex (blue) and the Alphafold model of SynAstaP (Uniprot P74615), indicating the nearly identical tunnel lining and different tunnel exits. Note that the side chain of the non-conserved Phe58 of P74615 clashes with the carotenoid polyene, that the non-conserved Gln56 of AstaPo1 contacting the polar groups of AXT is replaced by a shorter Thr57 residue in P74615, and that a conserved Trp79 of AstaPo1 is replaced by Phe80 in P74615 (the corresponding distances are shown in **a**). **d, e** The zeaxanthin-binding capacity of AstaPo1 (50 μM) and SynAstaP (Uniprot P74615) (500 μM) analyzed by SEC of proteins purified from *E. coli* cells synthesizing zeaxanthin. Note that SynAstaP has an extremely low extinction coefficient at 280 nm due to the total absence of Trp and the presence of only one Tyr residue, hence it was loaded in the tenfold higher concentration. The apparent $M_w$ of the peaks are shown. The absorbance spectra recorded during SEC runs and corresponding to the peak maxima are presented in the inserts. For SynAstaP, a similar spectrum of the protein obtained by mixing with ZEA in vitro prior to SEC is also presented. Absorbance maxima are indicated. The inserts in panel **d** also show the electrophoretic purity of the P74615 preparation used for the analysis and the appearance of the AstaPo1 and P74615 samples purified from ZEA-synthesizing *E. coli* cells. **f** Absorbance spectra of AstaPo1 WT and its mutant versions purified from ZEA-synthesizing *E. coli* cells under identical conditions. FWHM - full width at half maximum. Positions (nm) of the main spectral features are marked.

P74615, sequence identity with AstaPo1 ~36%). This protein, which we tentatively called SynAstaP, can, in principle, play a role similar to AstaPo1 because it has (i) a similar domain structure—an N-terminal tail with the signal peptide and a single FAS1 domain (residues 45–176), (ii) a similar predicted FAS1 domain fold (Fig. 5b), including the hydrophobic tunnel (Fig. 5c), which in AstaPo1 accommodates the carotenoid, and, finally, (iii) SynAstaP is also up-regulated in vivo by high light conditions[30]. Since the anticipated carotenoid-binding capacity of SynAstaP has not been tested yet[13], we produced the mature form of this protein (no signal peptide) in ZEA-synthesizing *E. coli* cells, purified it to homogeneity and characterized its properties in comparison with AstaPo1. SynAstaP could be well expressed and formed an incredibly soluble monomer (apparent SEC-derived $M_W$ ~22 kDa, electrophoretic mobility on sodium dodecyl sulfate-polyacrylamide gel electrophoresis ~17 kDa). However, under the conditions when AstaPo1 efficiently extracted ZEA and acquired the characteristic absorbance spectrum (Fig. 5d), SynAstaP could be obtained only as an apoprotein lacking any absorbance in the visible spectral region (Fig. 5e). Thus, under identical conditions, SynAstaP cannot bind ZEA like AstaPo1. Such a conclusion was confirmed upon mixing SynAstaP with pure carotenoids in vitro (Fig. 5e, insert) and hinted at the absence of some carotenoid-binding determinants or the presence of some hindrance to carotenoid binding in SynAstaP.

The structure of the FAS1 domain of SynAstaP predicted by Alphafold reveals a striking similarity with our AstaPo1 structure (Cα root-mean-square deviation (RMSD) of 1.8 Å) (Fig. 5b), with the hydrophobic tunnels of the two proteins being particularly similar (Fig. 5c). Among the 17 carotenoid-contacting positions found in AstaPo1 (Supplementary Fig. 11), the amino acid sequence of SynAstaP markedly differs by the $^{57}$TF$^{58}$ cluster replacing the $^{56}$QL$^{57}$ residues of AstaPo1, and also by the Phe residue replacing Trp79 (Fig. 5a). The neutral effect of the W79F replacement on the carotenoid-binding ability and the absorbance spectrum of AstaPo1 (Fig. 3h) likely qualifies the corresponding sequence difference, W79F, as non-crucial for carotenoid binding. At the same time, the Leu→Phe substitution at the generally conserved position 57 (Figs. 3f, 5a) introduces a bulkier residue on the tunnel periphery; at least one of the rotamers thereof can interfere with the carotenoid binding due to the possible clash

with the carotenoid polyene (Fig. 5c). Of note, a Phe in this position is present in as many as 88 out of 150 FAS1 homologs used for our Consurf analysis (Fig. 3f), while a Leu residue is present in only 23 of 150. The Q56T substitution could have also produced some effect on the carotenoid binding.

To provide further insight into the effect of the sequence differences between AstaPo1 and SynAstaP at positions 56 and 57, we prepared a double AstaPo1 mutant, Q56T/L57F, and the corresponding single mutants, and produced them individually in ZEA-synthesizing *E. coli* cells. Unexpectedly, mutant proteins extracted ZEA nearly as effectively as the wild-type AstaPo1 (reaching the high Vis/UV absorbance ratio for the purified protein of ~2.6–2.8). However, in the case of the Q56T/L57F and L57F mutants, the resulting absorbance spectrum experienced a significant distortion manifested by a ~3 nm blue shift, the appearance of the deep indent at 477 nm, and compaction of the whole Vis spectrum (FWHM decreased from 89 to 83 nm) (Fig. 5f). In turn, the Vis absorbance spectrum of the Q56T mutant (also containing an inadvertently emerged mutation A52V, which recapitulates Val of SynAstaP in this position, see Fig. 5a) was similar to that of the wild-type protein. These data indicated that while the L57F mutation did not abolish carotenoid binding by AstaPo1, it significantly affected the carotenoid microenvironment, most likely by distorting the shape of the carotenoid-binding tunnel. The Q56T substitution does not influence the carotenoid-binding mode as significantly.

The marked difference in carotenoid binding by AstaPo1 and SynAstaP is therefore explained by the remaining differences in their sequences, structures, and dynamics associated with the tentative carotenoid uptake process. The inability of SynAstaP to bind carotenoids strongly supports the idea that this function is a hallmark of only a selected set of FAS1 proteins.

## Discussion
In this work, we determined the first NMR structure of a carotenoid-protein complex, which, together with the biochemical data, revealed the peculiar mechanism of carotenoid embedment and explained the lack of AstaP specificity to carotenoids[15]. The determined structure is unique for carotenoid-binding proteins and supports the idea that, being found in many different organisms, carotenoid-binding proteins emanate from

the convergent evolution—their spatial organization, the mechanism of carotenoid uptake, and the ligand specificity are completely different.

For example, perhaps the earliest structurally described, β-crustacyanin is a 42-kDa heterodimer specifically binding two AXT molecules and controlling the coloration of lobster shells[26]. The Orange Carotenoid Protein (OCP) is a two-domain, 35 kDa photoswitching protein that binds a single ketocarotenoid molecule and regulates photoprotection in cyanobacteria[20,31,32]. The independently existing homologs of the N- and C-terminal domains of OCP, widely distributed among cyanobacteria, are carotenoid-binding proteins that participate in interprotein carotenoid transfer processes[33–35]. The structure of the 18 kDa C-terminal domain homolog (CTDH) has been determined by X-ray crystallography and NMR in the apoform only[21,36], but it is known to homodimerize upon carotenoid binding[28,35,37]. Being a rather specific ketocarotenoid binder, CTDH proved to be a robust carotenoid-delivery module which dissociates to monomers upon donating the carotenoid to other proteins or to biological membrane models[38]. In contrast, the homologs of the N-terminal domain of OCP, the so-called Helical Carotenoid Proteins, form rather small all-α helical ~20 kDa monomers, which are less specific and can bind at least deoxy-myxoxanthophyll, echinenone, canthaxanthin, and β-carotene[33]. Despite a broader ligand repertoire, HCP is a poorly efficient carotenoid extractor—it has always been purified along with the large excess of the apoform and required a CTDH homolog for the carotenoid delivery and maturation into the holoform[34,39]. The recently reported crystal structure of the 27 kDa Carotenoid-Binding Protein from silkworm[22], which determines the coloration of the silkworm cocoons[40], revealed a STARD3-like fold accommodating only part of the bound carotenoid molecule in the lipid-binding cavity[41,42]. This protein can bind a wide variety of carotenoids and can transfer them to other proteins, liposomes, and model cells, representing an attractive carotenoid-delivery module for various biotechnological and biomedical applications[43,44].

The AstaPo1 holoform structure determined here (Fig. 2) is dissimilar to any of these carotenoproteins. We show that its central domain has a FAS1-like fold, but the wide superfamily of FAS1-containing proteins has no common ligand-binding functions reported, not to mention that AstaP was discovered as the first FAS1 protein binding carotenoids[13]. Looking at the canonical FAS1 fold, it was nearly impossible to guess the exact location of the carotenoid-binding site a priori, especially since the homolog studied, AstaPo1, has rather long regions flanking the FAS1 domain, with the undefined structure and contribution to the carotenoid binding. We first localized the carotenoid-binding site to the FAS1 domain and showed that such miniaturized AstaPo1 (as small as 16 kDa) successfully matured into the holoform upon expression in carotenoid-producing E. coli strains, and is one of the smallest proteinaceous carotenoid-delivery modules currently known. Since the tailless apoform was fully insoluble, the tails, variable in amino acid composition and length in AstaP orthologs[16,17], are likely required for maintaining protein solubility in AstaPo1, which is especially important given that this protein is up-regulated under stress conditions and functions at high concentrations. We anticipate that the AstaPo1[41–190] variant can be shortened even slightly further without the loss of the carotenoid-carrying capacity.

On the background of other xanthophylls present, AstaPo1 isolated from native sources predominantly contains AXT[13,18]. Since the recombinant AstaPo1 demonstrates the apparently equal ability to bind xanthophylls CAN, ZEA, and AXT, including the ability to extract all of them from E. coli membranes upon production, we suggest that the AstaP-bound carotenoid content mostly reflects the accessible carotenoid pool; consequently, there should be additional factors dictating ligand specificity/accessibility in algal cells to override the remarkable promiscuity of AstaPo1. By controlling the pool of carotenoids loaded to AstaP, those yet undefined factors can, in theory, influence the light filtering property of AstaP (via adjusting its absorbance spectrum[15]) and its ROS quenching efficiency because of the different anti-oxidant properties of different carotenoids[45,46]. The existence of orange and pink AstaP homologs with different absorbance spectra is in favor of the assumption that the abundances of these photoprotective proteins can be adjusted upon some photo-acclimation processes, to filter out the most relevant irradiation.

Our NMR and mutagenesis data indicate that carotenoid binding involves a tunnel decorated by highly conserved hydrophobic amino acids, but its length is insufficient for accommodating the entire 30-Å carotenoid molecule. Thereby the carotenoid rings protrude from the globule and experience no apparent specificity restrictions, whereas the carotenoid polyene is stably fixed in the hydrophobic tunnel in a rather symmetrical conformation yielding no significant Vis-CD signal (Fig. 3). This binding mode significantly improves the thermal stability of AstaPo1 ($T_m$ increased from 60 °C by more than 15 °C). The only specific interactions with the carotenoid rings observed in the NMR structure are transient H-bonds involving the Gln56 side chain on the periphery of the tunnel. Although these H-bonds appear favoring for binding xanthophylls, they should have no role in the binding of β-carotene, and hence Gln56 is unlikely a strict carotenoid-binding determinant. It is tempting to suggest that Gln56 contributes to the preferences of AstaPo1 in xanthophyll versus β-carotene binding and this residue may take part in the carotenoid capture/extraction mechanism. The majority of AstaP orthologs with the reported or proposed carotenoid-binding function have this Gln (or its synonymous replacement by an Asn) (Fig. 5). Intriguingly, this position is not conserved at all in the context of the entire FAS1 superfamily, which is also the case for some other positions occupied by the carotenoid-contacting residues in our structure (e.g., positions 111 and 138 at one of the tunnel exits (Fig. 5)). In any case, the carotenoid-binding ability of any FAS1-containing protein besides AstaPs from Scenedesmaceae required experimental testing.

The analysis of covariation of presumable carotenoid-binding residues suggested that positions on the periphery of the tunnel can favor or disfavor the carotenoid-binding capacity, which could inform the sequence-based prediction of such function in AstaP homologs. However, the real carotenoid binding mechanism appears to be more complicated. Although the cyanobacterial homolog, SynAstaP, with a similar domain structure and an apparently unfavorable replacement of the Leu57 residue of AstaPo1 by a bulky Phe, could not bind carotenoids like AstaPo1, the L57F mutation in AstaPo1, alone or combined with another SynAstaP-mimicking mutation, Q56T, caused significant spectral changes indicating distorted, but not abolished carotenoid binding (Fig. 5). The effects of Q56T and A52V mutations on the carotenoid binding of AstaPo1 were rather neutral. The observed tolerance of AstaPo1 to the Q56T/L57F mutation and the retention of the carotenoid-binding ability are likely associated with the specific location of these residues in an unstructured five-residue loop connecting α1 and α2 helices, which can be rearranged upon the mutation into a conformation still capable of accommodating the carotenoid, yet in an altered, spectrally different binding mode (as observed in the experiment). In addition, the side chain of the introduced Thr56 residue can retain the ability to form H-bonds with the oxygenated groups on the carotenoid ring, which can explain the largely neutral effect of the Q56T substitution on ZEA binding by AstaPo1. Given the retained ZEA-binding ability of AstaPo1 mutants W79F, I176F, Q56T/A52V, L57F, and

Q56T/L57F, the complete inability of SynAstaP to bind carotenoids under identical conditions may indicate that not only tested amino acid replacements but also the global dynamics of the SynAstaP structure is different from AstaPo1 in not being suited for the carotenoid uptake mechanics depicted in Fig. 4.

Nevertheless, the inability of SynAstaP to bind carotenoids shown here supports the idea that carotenoid binding is pertinent to special FAS1 proteins. Since cell adhesion principles involving FAS1 domains are thought to have evolved at the earliest known stages of evolution[2], the carotenoid-binding function is likely an example of neofunctionalization of a subset of AstaP-like proteins within the green algae. Finding and experimentally validating carotenoid-binding orthologs of AstaPo1 warrant further interesting investigations.

## Methods

**Materials**. All-trans-astaxanthin (CAS Number: 472-61-7) was purchased from Sigma-Aldrich (USA). Absorbance spectra of carotenoids in organic solvents were registered on a Nanophotometer NP80 (Implen, Germany) using the following molar extinction coefficients: $125{,}000\ M^{-1}\ cm^{-1}$ for AXT at 482 nm in dimethyl sulfoxide (DMSO)[47], $145{,}000\ M^{-1}\ cm^{-1}$ for ZEA at 450 nm in methanol[48]. Chemicals were of the highest quality and purity available.

**Plasmid construction and mutagenesis**. AstaPo1 cDNA corresponding to residues 21–223 of the Uniprot S6BQ14 entry was codon-optimized for expression in *E. coli*, synthesized by Integrated DNA Technologies (Coralville, Iowa, USA) and cloned into the pET28-His-3C vector (kanamycin resistance) using the *NdeI* and *XhoI* restriction sites. SynAstaP cDNA corresponding to residues 27–180 of the Uniprot P74615 entry was codon-optimized for expression in *E. coli*, synthesized by Kloning Fasiliti (Moscow, Russia), and cloned as described above.

AstaPo1 truncated variants (AstaPo1$^{21-190}$, AstaPo1$^{41-223}$, and AstaPo1$^{41-190}$) or mutants W79F and I176F were obtained using Q5 (NEB) polymerase by the megaprimer PCR method. The single mutants, Q56T and L57F, were obtained using Phusion polymerase by the megaprimer PCR or the assembly PCR[49], respectively. The Q56T/L57F mutant was constructed by QuikChange™ site-directed mutagenesis method using Phusion polymerase according to ref. [50]. Primers are listed in Supplementary Table 1. The Q56T mutant contained an inadvertent mutation, A52V, which mimicked the Val residue of SynAstaP in this position and, therefore, was used for the analysis. The same vector and restriction sites as above were used for the mutants. The N terminus of all proteins contained extra residues GPHM after 3 C cleavage step. All resulting constructs were verified by DNA sequencing (Evrogen, Moscow, Russia).

**Protein production and sample preparation**. The wild-type AstaPo1, its mutant constructs, and SynAstaP were transformed into BL21(DE3) *E. coli* cells for expression of the apoforms. For the production of carotenoid-bound holoforms, plasmids described above were transformed into either BL21(DE3) carrying pACCAR25ΔcrtX plasmid (chloramphenicol resistance, ZEA biosynthesis), or BL21(DE3) containing pAC-CAR16ΔcrtX (chloramphenicol resistance, β-carotene biosynthesis) and the pBAD plasmid (ampicillin resistance, CAN biosynthesis). Details on the pathway of carotenoid biosynthesis in our system are described in ref. [15].

Expression cultures were grown in LB medium till $OD_{600} = 0.6$ at 37 °C and induced by 0.05–0.2 mM isopropyl-β-thiogalactoside (IPTG) for 24 h at 25 °C. In the case of wild-type holoforms, expression was carried at 30 °C and for CAN biosynthesis addition of 0.02% L-arabinose was also required. For expression of AstaPo1 mutant holoforms temperature was decreased to 25 °C.

All recombinant proteins were purified using the combination of subtractive immobilized metal-affinity and size-exclusion chromatography to electrophoretic homogeneity. Pure proteins were aliquoted and stored frozen at −80 °C. Protein concentrations were determined on a Nanophotometer NP80 (Implen) using the extinction coefficients given in Supplementary Table 2. For AstaP holoforms, protein extinction coefficients were used after subtraction of the carotenoid contribution into absorbance at 280 nm determined earlier, which are $A_{463}/6$ for ZEA and $A_{479}/7.4$ for CAN[15]. The expected Vis/UV absorbance ratios for AstaPo1 complexes with ZEA are listed in Supplementary Table 2. The carotenoid content of holoforms was analyzed by acetone-hexane extraction of carotenoids followed by thin-layer chromatography[43].

For NMR, the His$_6$-tagged AstaPo1 was expressed in *E. coli* BL21(DE3) cells. Overnight grown *E. coli* were diluted till $OD_{600} = 0.005$ in 2 L of fresh M9 minimal salt medium supplemented with Traces of metals (1:10000 v:v[51]) and containing 50 mg/L kanamycin. For isotopic labeling, $^{15}NH_4Cl$ and $^{13}C$ D-glucose were used. Cultures were grown to $OD_{600} = 0.6$ at 28 °C and then protein expression was induced by the addition of 0.1 mM IPTG for 24 h incubation at 16 °C.

Cells containing the overexpressed AstaPo1 were harvested at 7000×*g*, 4 °C for 5 min, and resuspended in 120 mL of lysis buffer (0.1 M Tris pH 7.5; 1 M NaCl; 1 mM β-mercaptoethanol (βME); 10 mM imidazole; 10% glycerol; 1% Triton X-100, and 0.2 mM phenylmethylsulfonyl fluoride). Resuspended cells were disrupted by

sonication for 30 cycles of 20 s sonication and 2.5 min of resting and the cell lysate was spun at 14,000×*g*, 4 °C for 1 h. The pellet was discarded and the supernatant was passed through a 0.22-μm filter (Millipore) and purified by affinity chromatography using a 5 mL Ni-NTA (Qiagen) column. Immobilized AstaPo1 was washed thrice, first with buffer containing 0.1 M Tris pH 7.5; 1 M NaCl; 1 mM βME; 0.5% Triton X-100, and 10 mM imidazole, next with the buffer containing 0.1 M Tris pH 7.5; 1 M NaCl; 1 mM βME, and 40 mM imidazole and final washing step with buffer containing 0.1 M Tris pH 7.5; 1 M NaCl; 1 mM βME, and 80 mM imidazole. His-tagged AstaPo1 was eluted with a buffer containing 0.1 M Tris pH 7.5; 1 M NaCl; 1 mM βME, and 500 mM imidazole. Eluted AstaPo1 was dialyzed overnight against 100x volume of 50 mM Tris buffer pH 7.5 containing 150 mM NaCl, in dialyzer D-Tube™ Dialyzer Mega, molecular weight cutoff 3.5 kDa (Merck) at 4 °C.

Overnight incubation with 3 C protease from human rhinovirus was used to remove the His$_6$-tag of AstaPo1. Next, the sample was centrifuged at 25,000×*g*, 4 °C for 1 h and purified by affinity chromatography using a 5 ml Ni-NTA column (Qiagen). Flow through with pure AstaPo1 was collected and dialyzed overnight against the buffer containing 50 mM Tris pH 7.6, 0.15 M NaCl in dialyzer D-Tube™ Dialyzer Mega, molecular weight cutoff 3.5 kDa (Merck) at 4 °C. Pure AstaPo1 was used immediately or stored at −20 °C.

To assemble the AstaPo1(AXT) complex, we mixed AXT in DMSO (1 mg mL$^{-1}$) with purified AstaPo1 at the 4.3-fold excess and incubated at moderate mixing, room temperature for 1 h, light-protected. DMSO contents in the solution were 10–12%. All protein-free AXT was removed by centrifugation at 25,000×*g*, 4 °C for 1 h. The supernatant was dialyzed overnight against the buffer containing 50 mM Tris pH 7.5 and 150 mM NaCl at 4 °C to remove the DMSO. The AstaPo1(AXT) complex was concentrated by an Amicon concentrator with 3.5 kDa molecular weight cutoff (Merck) to a concentration of 7 mg mL$^{-1}$ (280 μM).

**Analytical size-exclusion spectrochromatography**. SEC with continuous diode-array detection in the 240–850 nm range (recorded with 1 nm steps (4 nm slit width) and a 5 Hz frequency) was applied to study the apparent $M_w$ and absorbance spectra of the protein holoforms. Samples (50 μl) were loaded on a Superdex 200 Increase 5/150 column (GE Healthcare, Chicago, Illinois, USA) pre-equilibrated with a 20 mM Tris-HCl buffer, pH 7.5, containing 150 mM NaCl, 3 mM NaN$_3$, 5 mM βME, and operated using a Varian ProStar 335 system (Varian Inc., Melbourne, Australia). Diode-array data were converted into csv files using a custom-built Python script and processed into contour plots using Origin 9.0 (Originlab, Northampton, MA, USA).

**Circular dichroism**. AstaPo1 (0.5 mg mL$^{-1}$, 23 μM, Vis/UV absorbance ratio 2.54) or its truncated AstaPo1$^{41-190}$ mutant (0.42 mg mL$^{-1}$, 26.5 μM, Vis/UV absorbance ratio 3.35) were dialyzed overnight against 20 mM Na-phosphate buffer pH 7.05 and centrifuged for 10 min at 4 °C and 14,200×*g* before measurements. Visible/UV CD spectra were recorded at 20 °C in the range of 190–650 nm at a rate of 0.4 nm s$^{-1}$ with 1.0 nm steps in 0.1 cm quartz cuvette on a Chirascan circular dichroism spectrometer (Applied Photophysics) equipped with a temperature controller. The raw spectrum was buffer-subtracted before the presentation. Free ZEA (12 μM) in methanol was measured for reference. ZEA concentration in methanol was determined spectrophotometrically. The Vis/UV CD spectrum of the BmCBP(ZEA) complex (0.5 mg mL$^{-1}$, 18.5 μM, Vis/UV absorbance ratio 1.42) was taken from previous work[22]. For normalization, all CD spectra were scaled so that the corresponding absorbance spectra in the visible range were of the same amplitude.

To compare the secondary structures of the apo- and holoform of AstaPo1, we used the experimentally determined far UV CD spectrum of AstaPo1(apo)[15] and calculated the far UV CD spectrum of the AstaPo1(AXT) complex based on the NMR structures. To this end, we first calculated the CD spectra for each of the twenty NMR models in PDBMD2CD[29] and then scaled the average CD spectrum to that of the apoform for comparison.

**Fluorescence spectroscopy**. Steady-state Trp fluorescence emission spectra of 6 μM AstaPo1(apo), AstaPo1(ZEA) or AstaPo1$^{41-190}$(ZEA) were recorded upon excitation at 297 nm at 20 °C in the range of 305–450 nm at the rate of 30 nm min$^{-1}$ on a Cary Eclipse spectrofluorometer (Varian) in the 20 mM HEPES buffer (pH 7.7) containing 150 mM NaCl. The spectrum of the buffer was subtracted from the protein spectra before the presentation. The excitation and emission slits width was 5 nm. Indicated AstaPo1 concentrations are protein concentrations corrected for ZEA absorbance at 280 nm.

**Differential scanning calorimetry**. The AstaPo1 apoform (1.73 mg mL$^{-1}$, 80 μM) or its CAN- (1.66 mg mL$^{-1}$, 77 μM) or ZEA-bound (1.83 mg mL$^{-1}$, 85 μM) forms, or the truncated AstaPo1$^{41-190}$(ZEA) complex (1.3 mg mL$^{-1}$, 82 μM) were dialyzed overnight against a 20 mM Na-phosphate buffer (pH 7.0) and subjected to DSC on a VP-capillary DSC (Malvern) at a heating rate of 1 °C per min. Thermograms were processed using Origin Pro 8.0 and transition temperature ($T_m$) was determined from the maximum of the thermal transition.

**Small-angle X-ray scattering**. SAXS data ($I(s)$ versus $s$, where $s = 4\pi\sin\theta/\lambda$, $2\theta$ is the scattering angle and $\lambda = 1$ Å) from AstaPo1(CAN) (2.54 mg mL$^{-1}$) or AstaPo1(ZEA) (7.05 mg mL$^{-1}$) were measured at 20 °C at the BM29 beamline (ESRF,

Grenoble, France) using a Pilatus 2 M detector in a batch mode. The buffer included 20 mM Tris-HCl, pH 7.6, and 150 mM NaCl. The series of frames for each sample revealed no significant radiation damage. Six statistically matching frames for each protein sample were averaged and buffer-subtracted to produce the SAXS profile for further analysis of the SAXS-derived structural parameters, which are listed in Table 2. The SAXS data were processed and analyzed in PRIMUS[52]. CRYSOL[53] was used for calculating theoretical SAXS profiles, fitting, and validation. To assess the quality and uniqueness of the approximation of the SAXS data by NMR structure models, alternative models based on the previously determined crystal structures of other carotenoproteins were used.

**NMR spectroscopy.** NMR spectra were recorded using the Bruker Avance III 600 and 800 MHz spectrometers, both equipped with triple-resonance cryogenic probes. Spectra were acquired at two temperatures: 25 and 40 °C. Backbone NMR chemical shift assignment of AstaPo1, in both apo and holoforms, was performed using the conventional triple-resonance approach[54], signals of aliphatic side chains were assigned based on 3D hCCH- and HcCH-total correlation spectroscopy (TOCSY) experiments and aromatic residues were assigned using the HCCH-COSY[55], (HB)CB(CGCCCar)Har[56], and 3D $^1$H,$^{15}$N-NOESY-HSQC spectra. The complete sets of NMR experiments were recorded for both temperatures. In addition, to track the protein signals, we recorded the 2D NMR spectra at 30 and 35 °C. All the triple-resonance spectra were acquired using the BEST-TROSY pulse sequences[57], and most of the 3D NMR spectra were recorded in a non-uniformly sampled regime and were processed using the qMDD 2.5 software[58]. For AstaPo1 in the apo state, the assignment of the protein in complex with AXT was used as an additional source of data. $^3J_{CCo}$ and $^3J_{NCo}$ couplings were measured from the spin-echo difference constant-time HSQC spectra[59,60]. $^3J_{NH\beta}$ couplings were derived from the cross-peak intensities in a J-quantitative 3D HNHB experiment[61]. To assign the chemical shift of AXT, we used the $^{13}$C,$^{15}$N-double-filtered 2D NOESY, and TOCSY experiments[23]. Intramolecular distances were measured in 3D $^1$H,$^{15}$N-NOESY-HSQC, and $^1$H,$^{13}$C-NOESY-HSQC. Intermolecular contacts were observed directly using the 3D $^{13}$C,$^{15}$N-filtered,$^{13}$C-edited-NOESY-HSQC[62].

Spatial structures were calculated using the automated procedure as implemented in CYANA version 3.9.13[63]. The intermolecular NOESY peak list was assigned manually. The dihedral restraints ($\varphi$, $\chi_1$) were obtained from the manual analysis of J-couplings and chemical shifts in TALOS-N software[64]. CYANA format library for the ligand was based on a library for the structure of AXT in complex with β-crustacyanin[26]. This library was precalculated by the CYLIB algorithm[27]. Values of dihedral angles were set according to a quantum chemical study[24]. Dihedral angles of the ligand were restrained in the range (177;183) for most of the double bonds, the angle C5-C6-C7-C8 was restrained in the range (−173;−167) for the s-trans conformation and (−43;−37 or +37;+43) for the s-cis conformation. MOLMOL[65] and PyMOL software (Schrödinger LLC) was used for 3D visualization.

$^{15}$N longitudinal (T1), and transverse (T2) relaxation rate in the AstaPo1(AXT) complex were measured using the pseudo-3D HSQC-based experiments with varied relaxation delays at 25 °C using the Bruker Avance III 800 MHz spectrometer[66]. Spectra were recorded in the interleaved manner with the random order of data points, and a recycling delay of 3 s. Heteronuclear equilibrium $^1$H and $^{15}$N-NOE magnitudes were obtained using the $^1$H presaturation for 3 s during the recycling delay. Reference and NOE spectra were recorded in the interleaved mode. Relaxation parameters were analyzed using the model-free approach as implemented in the TENSOR2 software, assuming the isotropic rotation[67].

**Statistics and reproducibility.** Biochemical, spectroscopy, and spectro-chromatography data were collected at least three times for each of at least two separately purified protein batches obtained for each protein variant, and the most typical results were presented (no significant outliers were observed in any case). At least six SAXS frames were used to produce the profile used in further analysis and validation.

**Reporting summary.** Further information on research design is available in the Nature Portfolio Reporting Summary linked to this article.

## Data availability

The spatial structure of the AstaPo1(AXT) complex and NMR chemical shifts were deposited to the Protein Data Bank under access code 8C18 and to the BMRB database under access code 34781. The SAXS data for AstaPo1(ZEA) and AstaPo1(CAN) and the fits were deposited to the SASBDB under accession codes SASDRG8 and SASDRH8, respectively. The uncropped version of the gel in Fig. 5e is presented as Supplementary Fig. 12. Raw data behind the graphs are presented in Supplementary Data 1 (Microsoft Excel format). All other data are available from the corresponding authors upon request.

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

## Acknowledgements

The authors thank Dr. Anton Popov for help with SAXS data collection at the BM29 beamline (ESRF, Grenoble, France; experiment session data https://doi.org/10.15151/ESRF-ES-496408264). The study was supported by the Ministry of Science and Higher Education of the Russian Federation in the framework of Agreement no. 075-15-2021-1354 (07.10.2021). CD measurements were done at the Shared-Access Equipment Centre "Industrial Biotechnology" of the Federal Research Center "Fundamentals of Bio-technology" of the Russian Academy of Sciences. NMR studies were supported by the Grants Council of the President of the Russian Federation (grant МД-2834.2022.1.4).

## Author contributions

F.D.K. determined the NMR structure and prepared figures. Y.B.S. performed cloning and biochemical experiments and analyzed data. D.A.L. performed cloning and bio-chemical experiments. N.A.E. performed cloning and discussed the results. A.G.S. per-formed protein expression and purification. S.Y.K. performed differential scanning calorimetry measurements. E.G.M. obtained AXT and discussed the results. S.A.G. supervised protein production for NMR studies. K.S.M. conceived studies, analyzed NMR data, wrote the paper, acquired funding, and supervised research. N.N.S. conceived and designed studies, analyzed SAXS and biochemical data, wrote the paper, prepared figures, acquired funding, and supervised research.

## Competing interests

The authors declare no competing interests.
