## [Peer Review File · Communications Biology]

Reviewers' comments:

Reviewer #1 (Remarks to the Author, also attached):

The manuscript of Kornilov et al. aims to biochemically and structurally characterize a novel carotenoid-binding protein from the FAS1 family extracted from green algae. The manuscript provides a concise overview delineating the prevalence and role of the FAS1 superfamily in cell adhesion, detailing the structural motifs which join the members under this familial label.

The recently discovered protein belonging to this family, AstaPo1, was first described in its native condition to be an astaxanthin-binding protein, though a structural model was needed to describe the binding mechanism, and how this allows for ligand heterogeneity. The authors note that the truncated, yet functional, AstaPo1 that they are able to produce is 16kDa, which is the smallest proteinaceous carotenoid delivery molecule, making it a valuable candidate for carotenoid solubilization and targeted delivery. Non-specific carotenoid binding also emphasizes its potential for further utilization.

This study employs the use of several techniques to first localize the binding site and then structurally model it, offering an explanation as to how the different carotenoids (such as astaxanthin, zeaxanthin, canthaxanthin, beta-carotene) are able to bind to the protein. The experimental data on show constructs a convincing narrative that lets the reader understand the stepwise rationale employed to make assertions about structure from biochemical techniques. We commend the authors for the completeness of their work and for their carefulness in not overstating results.

However, there are some issues that should be addressed before the referee can recommend publication.

- Lines 108-110: "... the preprotein ... was accumulated on the cell periphery in response to hyperinsolation and osmotic stress, to perform a photoprotective role": what photoprotective role are the authors referring to? Without a clear function, the AstaPo1 protein is just a curiosity, so we encourage the authors to actually explain why their protein is important and why general readers should care. What is the function? Is the protein essential? Is a phenotype associated to its absence or knockout? Was the protein demonstrated to be a carotenoid transporter? to the exterior or the cell? with what purpose?

- Lines 168-170: "This suggests that the N- and C-terminal regions of AstaPo1 and carotenoid binding dramatically improve protein solubility.". So would that be their 'function'? What is generally the role of these N and C terminal domains in the other FAS1 proteins?

- Lines 173-174: "Given the tiny size of this water-soluble protein (16 kDa), it may be promising for various biomedical applications". Lines 521-523: "This protein can bind a wide variety of carotenoids and can transfer them to other proteins and liposomes, representing an attractive carotenoid-delivery module for various biotechnological and biomedical applications". Can the authors elaborate on the envisioned applications? Is their idea to deliver protein-bound carotenoid to humans? To what benefit?

- Lines 199-200: "To investigate the structure of the AstaPo1 complex with its native carotenoid, AXT, we took advantage of solution NMR spectroscopy.". Did the authors attempt to crystallize the protein? Given the high T_m (all the more with AXT bound) and small size, one would expect that this protein readily crystalizes – at least when devoid of the highly-dynamic N and C-terminal loops.

- Lines 200-201: "We synthesized the 13C/15N-labeled AstaPo1 apoprotein and complexed it with the unlabeled chemically pure AXT." Why didn't the authors directly study the holoprotein produced in E.

coli since they can control the carotenoid production in their cell and ensure presence of either CAN or ZEA or AXT?

- Section 2.5 (pages 12-14): This part of the paper reads as the weakest, as the importance of residues claimed to be central to carotenoid binding/uptake/stabilization is not supported by mutagenesis data. Sure, I176 and W79 have been mutated to Phe, resulting in reduced uptake of carotenoid and no effect, respectively, but L111 and A138 (proposed to have served the neo-evolution of Fas1 into AstaPo1 by enabling access to the hydrophobic tunnel), Q56 (proposed to partake in the stabilization of ketocarotenoid) and L57 (proposed to be essential to enable binding of carotenoid) have been left untouched, leaving open the burning questions raised by this Section. For example, would a 56-TF-57 -> 56-QL-57 replacement make P74615 capable of binding carotenoid (or reversely, would 56-QL-57 -> 56-TF-57 replacement abolish carotenoid binding by AstaPo1)? Likewise, would replacement of AstaPo1 A138 and V111 by a leucine and lysine, respectively, reduce/abolish carotenoid binding in AstaPo1 (or reversely, would replacement of P74615 V138 and K111 by an alanine and valine, endow it with carotenoid binding capability)? Last, would replacement of Q56 by a serine/alanine reduce/increase ketocarotenoid/carotene binding specificity?

- Lines 454-456: "This example supports the idea that it was variations of the peripheral residues at the exits of the hydrophobic tunnel that may have determined the newly acquired ability of some FAS1 members to bind ligands". That is a tantalizing hypothesis... Would it mean that this hydrophobic tunnel could serve binding of other ligands in other Fas1 proteins? Are there suggestions the authors would/could make in that respect?

- Line 552-555: "The analysis of covariation of such residues revealed that positions on the periphery of the tunnel can favor or instead disfavor the carotenoid-binding capacity, which informs the sequence-based prediction of such function in AstaP homologs". In the present version of this manuscript, this can only be considered as a suggestion, not a revelation/demonstration, since no experiment is presented that actually proves the point. A 56-QL-57 mutant of P74615 or a 56-QF-57 mutant of AstaPo1 could allow for it.

Also, there are some experimental choices which are left unclear.

- It is alluded to that the functionalization of AstaPo1 with zeaxanthin and canthaxanthin are favored experimentally due to the efficiency of their recombinant constitution in *E. coli*, but it isn't explained why zeaxanthin is favored over canthaxanthin during the localization of the binding site.

- It is also not explained why it is then necessary to conduct the NMR experiments using the native astaxanthin carotenoid. Even though it is inferred from the results that xanthophyll heterogeneity is not expected to change the protein structure, the referee thinks it important to explicitly state the reasons as to why certain carotenoids are used over others between different experiments for clarity.

- The discussion nicely summarizes what has been found as a result of this study and makes sure not to assert that which has not been seen from the experimental data, which while appropriate, could leave an unclear impression on the reader regarding biological relevance. This section does a good job of inferring potential from other carotenoid-binding proteins, but how does this relate the function of AstaPo1 as originally detailed? Can anything yet be said about how the structure of the protein-pigment complex, or ligand promiscuity, would benefit the cell during osmotic stress?

Last, but not least:

- It is not made clear during the main-text of the paper, outside of the references, what species of green algae AstaPo1 was isolated from, particularly the species that has been characterized here, other than the reference to the Scenedesmaceae family.

- Figure 1F insert: the photograph showing AstaPo1(ZEA) next to AstaPo1(CAN) would benefit from an additional absorption spectrum of AstaPo1(CAN) in UV/Vis. The insert is supposed to show the optical differences between the two variants, but one could also assume that this is due to a difference in concentration, especially as these two variants are said to have been used in SAXS at different concentrations (albeit with AstaPo1(CAN) more concentrated than Astapo1(ZEA)).

- A follow up question is as to whether or not functionalization by a different carotenoid would affect the biological function, notably the 'photoprotective' response to hyperinsolation?

- DSC should be added to the list of abbreviations – or spelled out fully. Given the high amount of already used abbreviations, the second option could be preferable.

Reviewer #2 (Remarks to the Author):

The presented paper by Kornilov et al. addresses an interesting question, how AstaPo1 binds the carotenoid astaxanthin. The authors performed a series of adequate measurements and determined the structure of AstaPo1 using NMR spectroscopy, they validated the structure by Small Angle Scattering X-ray Spectroscopy and also used other biophysical methods to characterize the binding properties of AstaPo1.

The paper is logical and well written and I have only minor questions.

1. The authors mention in Figure 1E that the tryptophan fluorescence is strongly quenched when astaxanthin (AXT) binds, and they find as as proof of binding. I accept this, but they should mention by what mechanism is the fluorescence quenched. The spectra shown in Fig1E makes clear that the tryptophan is not buried or hidden after binding, but the mechanism should be mentioned.
2. In the case of Fig2D the SAXS data shows a nice agreement between the predicted and measured SAXS curve. It is not totally relevant in this paper but why the OCP fit is so off?
3. The authors mention many times the strong binding affinity of AXT in different forms of the protein. It would be (or would have been) nice to add some Kd numbers. It would be interesting to know the Kd of the I176F mutant compared tot he WT. As they used fluorescence it would be just one step to measured the Kd of AXT binding.
4. The authors made the W79F mutant, and in this case one could see a small shift in the spectra. Do the authors expect that the W79 has a functional role in AstaPo1 ? If yes, what would be that role?
5. The authors proved that AstaPo1 binds carotenoids but what would be the final functional role of it, what would be the photochemistry ?

Reviewer #3 (Remarks to the Author):

The paper by Kornilov et al, describes the structural basis of an ancient protein FAS1 binding to carotenoid. Specially the authors determined the NMR structure of AstaPo1 and describes its mechanism of interaction with carotenoids. In addition using mutagenesis, the authors pinpoint the region on in AstaPo1 responsible for interaction and also showed that Trp79 rest in the interaction region. Further, the authors explain the basis of the unspecificity of AstaPo1 to their interaction with carotenoids.

I find the work quite thorough and I think the authors have used a wide range of techniques to support their finding. However, I think there are still a few things the authors will have to address to

make their finding adequately convincing for publications

Major comments

Did the authors performed a temperature gradient 1H-15N HSQC experiment or assignment experiment at 40°C before detecting the split signals? (S3). Will be interesting to see an overlay of the signals at 25 deg compared those at 40 deg. In addition, how many signals are present at 40 deg compared to those at 25 deg. How did the authors get the assignment of the residues at 40 deg if they did not do gradual temperature titration?

The RMSD of all 20 conformers of the AstaPo2/AXT complex is very small for a protein of this size and with so few restrains per residue. The very low CYANA target function also means that most restraints were satisfied. My concern here is the weight per restraints seems to be stronger than what it is given so few intermolecular NOEs. Can the authors explain how they reached this quality of structure with so few NOEs per residue? Based on the number of long range NOEs (460) , there is average of 2.2 long range NOES for every residues for a 207 amino acid. This can't give a structure well defined as presented

Page 10. The authors state without evidence of the apparent flexibility of the α 2-b1 hinge loop. They referred to figure 2B and S7. There is no evidence in the region of apparent flexibility. The NMR relaxation data is several similar across the protein chain. There are 2-3 residues without data but evidence of flexibility is not indicated. Can the authors provide evidence of this?

In the same paragraph as above the authors went further to explain that α 1- α 2 helices likely samples some conformational space without evidence. Would be interesting to see how the authors arrived at these conclusions. Can the author provide NMR relaxation data of the free AsPo1 and compare this with the complex form to substantiate their claim?

Did the authors also compared the NMR relaxation parameters of AstaPo1/AXT with those of AstaPo1/ZEA? Will be interesting to see how well this agrees with the SAXS data

Minor comments

Page 6. The authors used the word synthesised to refer to E. coli expressed protein. While this technically correct it is a bit confusing as proteins produced in E. Coli is generally referred to as "expressed" in protein science papers. I will suggest the authors change this throughout the paper to stick with the convention.

The authors should mentioned the concentrations of the AstaPo1/AXT used for the structure determination. The concentration mentioned on page 6, is it the concentration of AstaPo1 alone or a 50/50 mix?

Page 6. Will be nice to state which residues corresponds to the alpha 3 helix
Did the authors also compared the NMR relaxation parameters of AstaPo1/AXT with those of AstaPo1/ZEA? Will be interesting to see how well this agrees with the SAXS data.

Page 10. There is no S10 as the author referenced

Page 12. S11. There is no tertiary structure as the authors referenced in the text. The changes are mapped on the sequence but no tertiary structure.

Page 16 line 605. "Subtractive immobilized" what does the author mean by this?

Line 592 "synthesis" as mentioned above should be replaced to "express" and also in other parts of the text.

Line 742-748. The authors should mentioned the relaxation delays of the T1, T2. The delayed of the hetNOE seems to be 3s is rather too short which is only 2X the T1 times reported by the authors

Dear Reviewers

We would like to thank you for the fast and professional evaluation of our paper “Structural basis for the ligand promiscuity of the neofunctionalized, carotenoid-binding fasciclin domain protein AstaP” (COMMSBIO-23-0294-T).

Please find below our point-by-point responses (plain blue text) to the reviewer comments (bold black font).

On behalf of the authors

Nikolai N. Sluchanko

Reviewers' comments:

Reviewer #1 (Remarks to the Author):

The manuscript of Kornilov et al. aims to biochemically and structurally characterize a novel carotenoid-binding protein from the FAS1 family extracted from green algae. The manuscript provides a concise overview delineating the prevalence and role of the FAS1 superfamily in cell adhesion, detailing the structural motifs which join the members under this familial label.

The recently discovered protein belonging to this family, AstaPo1, was first described in its native condition to be an astaxanthin-binding protein, though a structural model was needed to describe the binding mechanism, and how this allows for ligand heterogeneity. The authors note that the truncated, yet functional, AstaPo1 that they are able to produce is 16kDa, which is the smallest proteinaceous carotenoid delivery molecule, making it a valuable candidate for carotenoid solubilization and targeted delivery. Non-specific carotenoid binding also emphasizes its potential for further utilization.

This study employs the use of several techniques to first localize the binding site and then structurally model it, offering an explanation as to how the different carotenoids (such as astaxanthin, zeaxanthin, canthaxanthin, beta-carotene) are able to bind to the protein. The experimental data on show constructs a convincing narrative that lets the reader understand the stepwise rationale employed to make assertions about structure from biochemical techniques. We commend the authors for the completeness of their work and for their carefulness in not overstating results.

Thank you!

However, there are some issues that should be addressed before the referee can recommend publication.

1.1. - Lines 108-110: “... the preprotein ... was accumulated on the cell periphery in response to hyperinsolation and osmotic stress, to perform a photoprotective role”: what photoprotective role are the authors referring to? Without a clear function, the AstaPo1 protein is just a curiosity, so we encourage the authors to actually explain why their protein is important and why general readers should care. What is the function? Is the protein essential? Is a phenotype associated to its absence or knockout? Was the protein demonstrated to be a carotenoid transporter? to the exterior or the cell? with what purpose?

The AstaP expression and astaxanthin synthesis are induced upon mixed stress conditions (high salt and intense light). Such stress, and also drought, often lead to ROS production, and antioxidant systems are necessary for survival. Microscopy studies showed that the orange layer outside of the plasma membrane becomes visible only after stress; AstaPo1 contains a signal peptide for protein translocation and extracellular localization, and its peripheral localization was observed under stress conditions (works of Prof Kawasaki).

It is thought that AstaP can filter out excess light (mainly, high-energy blue light) reducing photodamage and serve as an antioxidant preventing lipid oxidation in the plasma membrane. Unfortunately, AstaP gene knockout is yet to be done.

Our successful production of AstaP holoform in *E. coli* demonstrates that AstaP is capable of carotenoid extraction from the membranes without any supporting factors. Thus, due to equilibrium between carotenoid in protein matrix and in the lipid membrane, AstaP could be the reservoir of carotenoids for their delivery to membranes if they are oxidized during ROS production. Importantly, unlike OCP, AstaP does not seem to be a photoswitching protein which undergoes a light-induced conformational rearrangement, as its steady-state absorbance spectrum does not change much upon even very intense illumination (Slonimskiy et al FEBS J 2022).

Following the reviewer's recommendation, we supplemented the introduction of our paper with a more detailed description of the photoprotective role of AstaP reflecting the points above.

1.2. - Lines 168-170: “This suggests that the N- and C-terminal regions of AstaPo1 and carotenoid binding dramatically improve protein solubility.”. So would that be their ‘function’? What is generally the role of these N and C terminal domains in the other FAS1 proteins?

The length and amino acid content of N- and C-termini significantly differ even across AstaP-like proteins in algae from the *Scenedesma* family, not to mention FAS1 proteins in total. Because N- and C-termini are not that conservative, we cannot confidently generalize their potential role even in proteins closely related to AstaP. Nevertheless, the most logical interpretation based on a series of truncation mutants in our work, is that the disordered N and C tails present in some AstaP homologs improve protein solubility and stability. For example, the most truncated AstaPo1 variant we produced, deltaNC, was soluble only in complex with the

carotenoid, and we could not produce its apoform, which was mostly found in inclusion bodies. At the same time, the presence of C-term or both, C-term and N-term, enabled obtaining of the protein apoforms in the soluble state. Furthermore, N-terminus of AstaPo1 and some other homologs also contain N-glycosylations, which supports the idea that this protein segment is important for maintaining high protein stability and solubility under harsh conditions during osmotic stress characterized by the lack of water and thus increased local protein concentrations.

1.3. - Lines 173-174: “Given the tiny size of this water-soluble protein (16 kDa), it may be promising for various biomedical applications”. Lines 521-523: “This protein can bind a wide variety of carotenoids and can transfer them to other proteins and liposomes, representing an attractive carotenoid-delivery module for various biotechnological and biomedical applications”. Can the authors elaborate on the envisioned applications? Is their idea to deliver protein-bound carotenoid to humans? To what benefit?

Our works with this and other water-soluble carotenoid-binding proteins (e.g., *Bombyx mori* CBP) increasingly suggest that those proteins or their derivatives could indeed be used as small modules/containers for water-insoluble carotenoids, to help deliver the latter to various acceptor systems, be they other proteins (such as OCP, which is rendered photoactive upon the carotenoid transfer from AstaP and can thus be of use as an optogenetics tool - e.g., see Slonimskiy et al FEBS J 2022 and Piccinini et al Plant Physiol 2022), cell membranes of live cells (e.g. see Maksimov et al Antioxidants 2020; Semenov et al Antioxidants 2023). The protein-mediated carotenoid delivery to cell cultures avoids usage of organic solvents which inevitably hamper studies of carotenoid influence on human health. Instead of adding carotenoid solution (for example, in DMSO) to the model cell culture, we can apply a natural way to deal with carotenoids, using water-soluble carotenoid binding proteins. In the most recent work (Semenov et al Antioxidants 2023), we have shown that protein-mediated carotenoid delivery can counteract oxidative stress induced in the acceptor cells, and can alleviate deteriorating effects of lipofuscin in retinal pigment epithelium. Carotenoid delivery to model fibroblasts was shown to promote their growth (Slonimskiy et al IJBM 2022).

Direct delivery of the carotenoid-bound AstaPo1 to humans could potentially induce an immune response, but in the future one can think of humanization of the carotenoid-delivery systems/modules to make such a source of antioxidants more compatible. Among the beneficiary systems one can name that very retinal pigment epithelium, which is vulnerable to increased photoexcitation and photodamage (this function is exemplified by macula lutea in the retina, accumulating more than 1 mM of carotenoids in norm, but depleted in carotenoids upon age-related retina degeneration).

In addition, water-soluble carotenoid binding proteins may be used as an alternative to the classical way of carotenoid production and purification in batch cultures. Solubilized carotenoids could be exported from the cells in a continuous regime and then purified without any organics following best practices of green chemistry.

1.4. - Lines 199-200: “To investigate the structure of the AstaPo1 complex with its native carotenoid, AXT, we took advantage of solution NMR spectroscopy.”. Did the authors

attempt to crystallize the protein? Given the high T_m (all the more with AXT bound) and small size, one would expect that this protein readily crystallizes – at least when devoid of the highly-dynamic N and C-terminal loops.

This is a good point. Although there were indeed chances that AstaPo1 deltaNC would crystallize, we wanted to include also the termini in the structural analysis because one could not a priori exclude the involvement of their residues in carotenoid embedment, and the expected value of structural analysis for the full protein was much higher than only for its FAS1 domain. Nevertheless, we cannot exclude that future attempts could enable crystal structure determination of AstaP, although we note that a high T_m alone is a poor predictor of protein crystallizability.

1.5. - Lines 200-201: “We synthesized the 13C/15N-labeled AstaPo1 apoprotein and complexed it with the unlabeled chemically pure AXT.” Why didn’t the authors directly study the holoprotein produced in *E. coli* since they can control the carotenoid production in their cell and ensure presence of either CAN or ZEA or AXT?

This is an excellent notion, thanks for bringing that on. The point is that it is currently possible to get a homogeneously expressed ZEA (or CAN) bound protein in *E. coli*, while it is barely possible to achieve 100% AXT production in *E. coli* due to the presence of intermediates of its biosynthetic pathway. In our hands, it was possible to produce only a ZEA/AXT mixture with a different ratio between ZEA and AXT (CrtW ketolase activity was likely a limiting factor). As the result, AstaPo1 expressed in AXT producing *E. coli* extracted and bound both ZEA and AXT, and such ligand heterogeneity was unfavorable for structural studies focused on identifying protein-pigment interactions.

1.6. - Section 2.5 (pages 12-14): This part of the paper reads as the weakest, as the importance of residues claimed to be central to carotenoid binding/uptake/stabilization is not supported by mutagenesis data. Sure, I176 and W79 have been mutated to Phe, resulting in reduced uptake of carotenoid and no effect, respectively, but L111 and A138 (proposed to have served the neo-evolution of Fas1 into AstaPo1 by enabling access to the hydrophobic tunnel), Q56 (proposed to partake in the stabilization of ketocarotenoid) and L57 (proposed to be essential to enable binding of carotenoid) have been left untouched, leaving open the burning questions raised by this Section. For example, would a 56-TF-57 -> 56-QL-57 replacement make P74615 capable of binding carotenoid (or reversely, would 56-QL-57 -> 56-TF-57 replacement abolish carotenoid binding by AstaPo1)? Likewise, would replacement of AstaPo1 A138 and V111 by a leucine and lysine, respectively, reduce/abolish carotenoid binding in AstaPo1 (or reversely, would replacement of P74615 V138 and K111 by an alanine and valine, endow it with carotenoid binding capability)? Last, would replacement of Q56 by a serine/alanine reduce/increase ketocarotenoid/carotene binding specificity?

We would like to thank the reviewer for this important question. First, we note that some statements in this question are not fully correct. For example, we did not claim that the L111 and A138 residues served the neo-evolution of FAS1 into AstaP, instead just stated that those

residues are much more conserved in AstaP homologs than in FAS1 proteins in general, which supports the overall idea that not any FAS1 is tailored for carotenoid binding. Likewise, we cannot agree on the necessity and/or relevance of the V111K mutation (why K if P74615 also contains Val?) and the A138L mutation, which most likely be neutral (present in bona fide carotenoid binders - pink1 and pink2 AstaPs, see Fig. 5A). Second, our rich experience with site-directed mutagenesis of carotenoid-binding proteins (OCP, CTDH, BmCBP, STARD3, AstaP studied in our laboratory) tells us that mutations are not always effective for understanding the carotenoid-binding mechanism and can sometimes give unexpected results. For example, the a priori robust mutation I176F introduced in the middle of the carotenoid binding tunnel failed to abolish carotenoid binding, and only partially inhibited it (by ~40%), also causing severe changes to the absorbance spectrum (Fig. 3H). This effect is likely associated with the robustness of the evolved protein and that its structure can adjust to some of the introduced changes, which becomes even more probable in the case of rather flexible and dynamic proteins like AstaP.

With that caveat, we carefully considered the suggested mutations and agreed that the most interesting mutant would involve the combination of Q56T and L57F replacements in AstaPo1, to change the apparently auxiliary Q56 forming transient H bonds with the carotenoid by Thr found in P74615, and to introduce a bulky Phe (also found in P74615) instead of the conserved Leu in the beginning of the carotenoid tunnel. In fact this mutant was cloned, produced and analyzed already during the review process and we have now added the corresponding results to the revised version of the paper (Fig. 5F).

AstaPo1 Q56T/L57F double mutant was produced in the ZEA strain and turned to extract carotenoids almost as effectively as the WT AstaPo1. Nevertheless, the mutation was not neutral - we observed the dramatic distortion of the absorbance spectrum: 1) a ~3 nm blue shift, 2) the appearance of the deep indent at 477 nm, and 3) a compaction of the whole Vis spectrum (FWHM decreased from 89 to 83 nm) (Fig. 5F). These spectral signatures indicate a significant carotenoid torsion and underline the robustness of the carotenoid-binding mechanism by AstaPo1, which can tolerate many moderate point mutations carefully designed to prevent direct effects on the protein stability (for example, a L57K/E mutation would destroy the protein). We suggest that the retention of carotenoid binding by the AstaPo1 Q56T/L57F double mutant is explained by the location of the 56 and 57 positions in the flexible loop (depicted by magenta in the picture below) connecting the a1 and a2 helices of the jaw, which can be displaced to accommodate the inserted mutations. Moreover, the Thr residue inserted instead of Gln can still form H bonds with the carotenoid oxygen groups and thus can also be tolerated.

Revision Fig. 1. The superposition of the NMR structure of AstaPo1 (cyan) with the AlphaFold model of its Q56T/L57F mutant (green) showing the location of the 56 and 57 residues in the flexible loop (magenta), which can compensate for the introduction of the bulky residues by a small positional displacement from the carotenoid (shown by orange sticks). The retention of the H bonding interactions of the carotenoid ring with the sidechain of Thr56 is shown by a yellow dashed line.

Given the new data on the double mutant of AstaPo1, which we carefully analyzed and included in the revised version (see the new version of Fig. 5 below), we have reconsidered the statements concerning the role of 56 and 57 residues in discriminating carotenoid binding and nonbinding FAS1 proteins. While this did not affect conclusions and main points of our study that carotenoid binding is a neofunction of a subset of FAS1 proteins, we note that, obviously, finding and experimentally validating further carotenoid-binding orthologs of AstaPo1 will be required to further refine the carotenoid binding mechanism and understand its universality. We thank the reviewer again for his/her insightful question which allowed us to get new interesting data and improve the paper.

17 positions of AstaPo1 residues with >10 Å² changed SASA upon AXT binding:

magenta – identical/similar in Car binders, non-conserved in FAS1 superfamily

orange – non-conserved in Car binders, non-conserved in FAS1 superfamily

cyan – identical/similar in Car binders, conserved in FAS superfamily

New Fig. 5, rearranged panels B-E and new panel F.

1.7. - Lines 454-456: “This example supports the idea that it was variations of the peripheral residues at the exits of the hydrophobic tunnel that may have determined the newly acquired ability of some FAS1 members to bind ligands”. That is a tantalizing hypothesis... Would it mean that this hydrophobic tunnel could serve binding of other ligands in other Fas1 proteins? Are there suggestions the authors would/could make in that respect?

This is also a very good point. Our proposed structure-based ligand binding mechanism by AstaPo1 implies that any long enough hydrocarbon chain could be adopted in the hydrophobic tunnel. We are currently exploring this hypothesis and have acquired first confirmations, which if remaining consistent will become subject of the following publication(s). Likewise, it is also tempting to speculate that FAS1 containing proteins, frequently localized outside of the plasma membrane, could be receptors for various hydrophobic substances, however, it is yet to be tested.

1.8. - Line 552-555: “The analysis of covariation of such residues revealed that positions on the periphery of the tunnel can favor or instead disfavor the carotenoid-binding capacity, which informs the sequence-based prediction of such function in AstaP homologs”. In the present version of this manuscript, this can only be considered as a suggestion, not a revelation/demonstration, since no experiment is presented that actually proves the point. A 56-QL-57 mutant of P74615 or a 56-QF-57 mutant of AstaPo1 could allow for it.

As outlined above, we have obtained the double mutant of AstaPo1 with the simultaneous Q56T and L57F replacements, analyzed its ability to bind carotenoids and characterized its spectral properties in comparison with the WT protein. While we still observe ZEA binding to this new protein, the absorbance spectrum of such complex is substantially different from that of the WT AstaPo1, which indicates seriously changed carotenoid binding mode, potentially involving the distortion of the carotenoid binding tunnel and the conformation of the bound polyene. In light of these new data, we have rewritten the corresponding description by omitting the statements that the 56 and 57 positions are discriminators of carotenoid binding/nonbinding in FAS1 proteins.

1.9. Also, there are some experimental choices which are left unclear.

- It is alluded to that the functionalization of AstaPo1 with zeaxanthin and canthaxanthin are favored experimentally due to the efficiency of their recombinant constitution in *E. coli*, but it isn't explained why zeaxanthin is favored over canthaxanthin during the localization of the binding site.

The reason for our preference of the ZEA strain for localization of the carotenoid binding site was purely technical. The CrtZ enzyme in ZEA producing *E. coli* is so active that we could not detect anything but pure zeaxanthin. On the contrary, the CrtW enzyme in the CAN producing strain is much less active and hence both beta-carotene and canthaxanthin are accumulated. For the clarity of interpretations, we chose the strain where only one carotenoid was present. We believe that the same results could be obtained should we choose the CAN strain.

1.10. - It is also not explained why it is then necessary to conduct the NMR experiments using the native astaxanthin carotenoid. Even though it is inferred from the results that xanthophyll heterogeneity is not expected to change the protein structure, the referee thinks it important to explicitly state the reasons as to why certain carotenoids are used over others between different experiments for clarity.

Before the current study and our previous work it was thought that AstaPo1 is specific to astaxanthin as it is the predominant carotenoid in AstaP extracted from the native source (algae). We showed that AstaPo1 has a wide repertoire (Slonimskiy et al FEBS J 2022), however, we could not exclude that both hydroxy- and keto groups are involved in binding with the protein. AXT usage for NMR studies allowed us to make no preliminary assumptions about the binding mechanism. At the same time, any xanthophyll (and strain) could be used for biochemical/truncation analyses. These points we now clearly indicated in the main text.

1.11. - The discussion nicely summarizes what has been found as a result of this study and makes sure not to assert that which has not been seen from the experimental data, which while appropriate, could leave an unclear impression on the reader regarding biological relevance. This section does a good job of inferring potential from other carotenoid-binding proteins, but how does this relate the function of AstaPo1 as originally detailed? Can anything yet be said about how the structure of the protein-pigment complex, or ligand promiscuity, would benefit the cell during osmotic stress?

Thank you. Our study clearly shows that AstaP is a universal module for probably any carotenoid with the fixed size of hydrocarbon region (length of the tunnel). Any synthesized carotenoids could be accumulated and used against ROS during stress or as a light filter (although may be used for certain adjustments). Cells can modulate the properties of AstaP just by regulation of carotenoid biosynthesis (and/or regulating carotenoid accessibility to AstaP), which is a simple and elegant way. These ideas are now added to the discussion.

1.12. Last, but not least:

- It is not made clear during the main-text of the paper, outside of the references, what species of green algae AstaPo1 was isolated from, particularly the species that has been characterized here, other than the reference to the Scenedesmaceae family.

We are especially thankful for this point, which we overlooked. We now added the corresponding species to the required places and provided more description on the terminology orange/pink AstaPs to the introduction.

1.13. - Figure 1F insert: the photograph showing AstaPo1(ZEA) next to AstaPo1(CAN) would benefit from an additional absorption spectrum of AstaPo1(CAN) in UV/Vis. The insert is supposed to show the optical differences between the two variants, but one could also assume that this is due to a difference in concentration, especially as these two variants are said to have been used in SAXS at different concentrations (albeit with AstaPo1(CAN) more concentrated than Astapo1(ZEA)).

We showed the appearance of protein samples of AstaPo1 (ZEA) and AstaPo1 (CAN) at the same protein concentration for reference only. The detailed comparison of the two forms was done in previous work Slonimskiy et al FEBS J 2022, so we prefer not to repeat this in our current paper.

1.14. - A follow up question is as to whether or not functionalization by a different carotenoid would affect the biological function, notably the 'photoprotective' response to hyperinsolation?

Carotenoids differ by their antioxidant properties (AXT is the most potent antioxidant), so most probably changing the specificity of the carotenoid binding protein or the variety of synthesized carotenoids should affect its ROS quenching function, but less significantly, the light filtering effect of AstaP holoprotein, because of the similarity of the absorbance spectra with ZEA or CAN or AXT. Our structure implies that AstaPo1 carotenoid content follows the available carotenoid pool, which can indicate the presence of extra factors controlling the access of AstaP to specific carotenoids. We expanded this discussion in the main text of the revised version.

1.15. - DSC should be added to the list of abbreviations – or spelled out fully. Given the high amount of already used abbreviations, the second option could be preferable.

DONE

Reviewer #2 (Remarks to the Author):

The presented paper by Kornilov et al. addresses an interesting question, how AstaPo1 binds the carotenoid astaxhantin. The authors performed a series of adequate measurements and determined the structure of AstaPo1 using NMR spectroscopy, they validated the structure by Small Angle Scattering X-ray Spectroscopy and also used other biophysical methods to characterize the binding properties of AstaPo1. The paper is logical and well written and I have only minor questions.

Thank you very much!

2.1. The authors mention in Figure 1E that the tryptophan fluorescence is strongly quenched when astaxhantin (AXT) binds, and they find as as proof of binding. I accept this, but they should mention by what mechanism is the fluorescence quenched. The spectra shown in Fig1E makes clear that the tryptophan is not buried or hidden after binding, but the mechanism should be mentioned.

Most probably the mechanism of such quenching is a nonradiative energy transfer from the tryptophan residue to the carotenoid bound in the vicinity. This is now reflected in the corresponding subsection of results.

2.2. In the case of Fig2D the SAXS data shows a nice agreement between the predicted and measured SAXS curve. It is not totally relevant in this paper but why the OCP fit is so off?

The reason for that is simple: OCP is a 35 kDa two domain protein which has a completely different distribution of electron density in the molecule compared to AstaPo1, and hence its SAXS scattering profile is significantly different. Following best practices of the SAXS data analysis, OCP was used in validation here for comparison with AstaP, to give a perception of the discriminative power of the data in telling the difference between two small proteins of a comparable size (similar Rg values) but different 3D structure.

2.3. The authors mention many times the strong binding affinity of AXT in different forms of the protein. It would be (or would have been) nice to add some Kd numbers. It would be interesting to know the Kd of the I176F mutant compared to the WT. As they used fluorescence it would be just one step to measure the Kd of AXT binding.

Measurement of Kd for AstaPo1 and its mutant variants is complicated by the fact that generally carotenoids are insoluble in water solutions, which formally makes Kd close to zero, regardless of the water-soluble carotenoid binding protein or carotenoid type, and of the mechanism of binding. For instance, prolonged storage or exhaustive dialysis do not cause carotenoid dissociation from the protein. In other words, once formed, a carotenoid-protein complex would never dissociate because of the too high barrier for the carotenoid to leave the protein cavity/tunnel. The case of carotenoproteins deviates from the classical formalism for Kd determination in equilibrium.

For carotenoid-binding proteins one could technically measure the apparent Kd in the presence of a suitable detergent (with the assumption that it does not affect the binding and is not bound itself), but we believe that the results would be specific for the carotenoid vs detergent vs protein type, and therefore no good and universal comparison would be possible. We are in progress of elaborating a convenient means to compare the carotenoid binding efficiencies, but this is certainly outside the scope of the present work.

2.4. The authors made the W79F mutant, and in this case one could see a small shift in the spectra. Do the authors expect that the W79 has a functional role in AstaPo1? If yes, what would be that role?

According to our structural model, any bulky hydrophobic residue (W,F,Y) will fit into such a position of the carotenoid binding tunnel and will not disturb the carotenoid binding. The specific indolyl group of Trp79 apparently is not necessary for carotenoid binding and does not form direct chemical contacts such as H bonds to the carotenoid in the complex (but we cannot exclude its role in the process of the carotenoid uptake).

It is remarkable that a carotenoid spectrum in the protein is so sensitive to the surrounding residues and this could potentially be used for color tuning of carotenoprotein.

2.5. The authors proved that AstaPo1 binds carotenoids but what would be the final functional role of it, what would be the photochemistry ?

Previously (Kawasaki et al 2013 Plant Cell Physiol 54, 1027–40), it was shown that AstaPo1 (AXT) is capable of singlet oxygen quenching and its expression is highly induced during complex light-salt stress. Protein itself is accumulating outside of the algal cell due to the signal peptide for translocation. This leads to the conclusion that its functional role is most likely filtering the light and ROS quenching (both chemical and physical). Since carotenoids are characterized by extremely short lifetimes of excited states, their possible functional role in photoprotective proteins may be attributed to the conversion of electron excitation energy into heat. Thus, it is likely that cells can be shielded from solar radiation in the blue-green part of the spectrum and conversion of absorbed energy into heat might be useful for the reduction of the probability of formation of reactive oxygen species and photodamage of cellular structures.

Reviewer #3 (Remarks to the Author):

The paper by Kornilov et al, describes the structural basis of an ancient protein FAS1 binding to carotenoid. Specially the authors determined the NMR structure of AstaPo1 and describes its mechanism of interaction with carotenoids. In addition using mutagenesis, the authors pinpoint the region on in AstaPo1 responsible for interaction and also showed that Trp79 rest in the interaction region. Further, the authors explain the basis of the unspecificity of AstaPo1 to their interaction with carotenoids.

I find the work quite thorough and I think the authors have used a wide range of techniques to support their finding. However, I think there are still a few things the authors will have to address to make their finding adequately convincing for publications

Thank you!

Major comments

3.1. Did the authors performed a temperature gradient 1H-15N HSQC experiment or assignment experiment at 40oC before detecting the split signals? (S3). Will be interesting to see an overlay of the signals at 25 deg compared those at 40 deg. In addition, how many signals are present at 40 deg compared to those at 25 deg. How did the authors get the assignment of the residues at 40 deg if they did not do gradual temperature titration?

Indeed, we recorded the 15N and 13C HSQC experiments and X-filtered 1D spectra of the ligand at 25, 30, 35 and 40 degrees. We did the assignment mostly based on the triple resonance experiments: we have two sets of NMR spectra recorded, at 25 and 40 degrees.

Spectral quality at 25 and 40 degrees are very similar, splitting is clearly observed at both temperatures. The major difference was in the quality of the NOESY spectra - fewer signals were observed at 25 degrees, due to the effect of faster transverse relaxation. To make it clear, we described the assignment process in more detail in the experimental section paragraph 4.9, and also provide the overlay of ^1H - ^{15}N HSQCs here, for the purposes of peer review.

Revision Fig. 2. Signal splitting in ^1H , ^{15}N -HSQC spectra of AstaPo1/AXT complex recorded at 25°C (red color) and 40°C (blue color). Split signals are shown by numbered dashed rectangles and are additionally shown at high resolution on the right and left sides.

3.2. The RMSD of all 20 conformers of the AstaPo2/AXT complex is very small for a protein of this size and with so few restraints per residue. The very low CYANA target function also means that most restraints were satisfied. My concern here is the weight per restraints seems to be stronger than what it is given so few intermolecular NOEs. Can the authors explain how they reached this quality of structure with so few NOEs per residue? Based on the number of long range NOEs (460), there is average of 2.2 long range NOES for every residues for a 207 amino acid. This can't give a structure well defined as presented

First of all, we would like to point out here that we provide the RMSD values only for the elements of the secondary structure, excluding the loops and disordered terminal regions. Taking into account the whole FAS1 domain core, one would obtain the RMSD of ... for the protein backbone. Second, out of 203 residues in our protein constructs, 16 N-terminal residues and 20 C-terminal residues are unstructured and unrestrained, therefore, the provided number of distance restraints describe the structure of a 167-residue core. Apart from 460 long-range distances, we have 150 intermolecular contacts (which is a rather large amount), thus in total 610 long-range distances describe the structure of a 167-residue protein, which correspond to 3.65 long-range distance restraint per residue. Finally, apart from the distance restraints, we determined the backbone and sidechain dihedral angles for most of the residues, based on the J-couplings and chemical shifts, which also improves the quality of the structure dramatically compared to unrestrained dihedrals. To take into account this comment, we modified the statistical table, which now mentions the size of the structured protein core and provides the RMSD values for the whole structured core in addition to the previously reported value. We also stated explicitly that 150 intermolecular NOEs supplement the AstaPo1 NOEs.

3.3. Page 10. The authors state without evidence of the apparent flexibility of the α 2-b1 hinge loop. They referred to figure 2B and S7. There is no evidence in the region of apparent flexibility. The NMR relaxation data is several similar across the protein chain. There are 2-3 residues without data but evidence of flexibility is not indicated. Can the authors provide evidence of this?

When suggesting that the α 2-b1 loop is apparently flexible, we based on the increased RMSD for this region (Figure 2B) and slightly decreased order parameters, which can be clearly seen on Figure S8, which we forgot to cite in this context. Anyway, this was a hint that we used to put forward a hypothesis that this part of the protein may be mobile in the apo state. The hypothesis was then directly tested and proved by our study of the AstaPo1 apo state. We observe that the NMR signals of the whole α 1- α 2 jaw and the adjacent loops are either not observed or broadened dramatically in the NMR spectra, suggesting the presence of slow to intermediate conformational transitions in the us-ms timescale. These results are shown in Figures 4A and S9. To make our text clearer, we rewrote the first paragraph of 2.4 and now we do not start with the hypothesis, but base only on the data that we actually do have on the protein apo state.

3.4. In the same paragraph as above the authors went further to explain that α 1- α 2 helices likely samples some conformational space without evidence. Would be interesting to see how the authors arrived at these conclusions. Can the author provide NMR relaxation data of the free AsPo1 and compare this with the complex form to substantiate their claim?

As indicated above and in the paper, the NMR signals of the whole α 1- α 2 jaw and the adjacent loops are either not observed or broadened dramatically in the NMR spectra, suggesting the presence of slow to intermediate conformational transitions in the us-ms timescale. We did not manage to assign this part of the protein in the apo state, but as one can see, there are almost no unassigned cross-peaks left in the HSQC spectrum shown in the figure S9. Since the NMR assignment and the mere NMR signals of the most interesting part are absent, we doubt that

the NMR relaxation data for the assigned part would provide any understanding for the processes occurring in the "NMR-invisible" a1-a2 jaw. To make our evidence clearer, we rewrote the corresponding paragraph in the revised version of the text, avoiding speculative parts. In essence, although still being only a carefully formulated hypothesis, our insights into the carotenoid uptake mechanism of AstaP, consisting in the conformational change involving the movement of the a1-a2 jaw, is based on 1) the profound flexibility in the holoform structure of the linker between a2 and b1 connecting the whole jaw and the rest of the protein, 2) the "NMR invisibility" of the whole jaw in the apoform structure and 3) the comparison of our carotenoid-bound AstaPo1's FAS1 structure (closed) with the unliganded apoprotein structure of CupS (open). We believe that while our interpretations are carefully balanced by the available data used in the analysis, future studies can help refine the proposed hypothesis.

3.5. Did the authors also compared the NMR relaxation parameters of AstaPo1/AXT with those of AstaPo1/ZEA? Will be interesting to see how well this agrees with the SAXS data

AXT, the major native AstaPo1 ligand, is the only carotenoid that was investigated by NMR. Investigation of other carotenoids would require the chemical shift assignment (production of ¹³C/¹⁵N-labeled proteins). Taking into account the low yields of AstaPo1 and problems with protein stability at high concentrations (above as low as 7 mg/ml) as well as the modest potential insights provided by their analysis, studies of other carotenoid complexes would not be economically feasible. Besides, according to the structure obtained there is no reason to assume that other carotenoids would demonstrate any distinct modes of binding.

Minor comments

3.6. Page 6. The authors used the word synthesised to refer to E. coli expressed protein. While this technically correct it is a bit confusing as proteins produced in E. Coli is generally referred to as "expressed" in protein science papers. I will suggest the authors change this throughout the paper to stick with the convention.

Corrected.

3.7. The authors should mentioned the concentrations of the AstaPo1/AXT used for the structure determination. The concentration mentioned on page 6, is it the concentration of AstaPo1 alone or a 50/50 mix?

The 300 uM concentration refers to the protein and also describes the AstaPo1/AXT complex, since the protein and carotenoid exist in the ratio 1:1. The concentration and buffer conditions are also provided on page 17, paragraph "Protein synthesis and sample preparation" (280 uM). To disambiguate, we clarified the concentration on p. 6.

3.8. Page 6. Will be nice to state which residues corresponds to the alpha 3 helix

We specified the residues that comprise helices 6 and 3, that are mentioned in the text, in the revised version of the manuscript.

3.9. Did the authors also compared the NMR relaxation parameters of AstaPo1/AXT with those of AstaPo1/ZEA? Will be interesting to see how well this agrees with the SAXS data.

The identical question of this referee (3.5) was answered above.

3.10. Page 10. There is no S10 as the author referenced

Figures S10 and S11 were present on the last page of supplementary materials.

3.11. Page 12. S11. There is no tertiary structure as the authors referenced in the text. The changes are mapped on the sequence but no tertiary structure.

Thanks for pointing this out. In the figure, the changes in the solvent accessible surface area are shown along the protein sequence (and on top of the structural elements), to identify the residues experiencing the most pronounced changes upon the carotenoid binding. Their mapping on the tertiary structure of the protein is not directly depicted in the figure, but immediately follows from this analysis. We removed “tertiary” and left only “the primary and secondary structure”, also added the reference to Fig. 3F.

3.12. Page 16 line 605. “Subtractive imimobilized” what does the author mean by this?

In fact, the complete term is subtractive IMAC (immobilized metal-affinity chromatography), which designates a combination of immobilized metal affinity chromatography (IMAC) at the initial stage of the protein purification, subsequent cleavage of the hexahistidine tag, and the second round of IMAC to separate proteolyzed protein from the protease, tag and the fraction of undigested protein. The term is used rather often, hence we did not change anything in the text.

3.13. Line 592 “synthesis” as mentioned above should be replaced to “express” and also in other parts of the text.

OK

3.14. Line 742-748. The authors should mentioned the relaxation delays of the T1, T2. The delayed of the hetNOE seems to be 3s is rather too short which is only 2X the T1 times reported by the authors

The interscan delays for T1 and T2 measurements were also equal to 3s. We specified this in the Methods text as requested by the reviewer.

REVIEWERS' COMMENTS:

Reviewer #1 (Remarks to the Author):

The authors have addressed all of my comments.
I wish to them the best with the continuation of the publication process.

Reviewer #3 (Remarks to the Author):

Review of revised manuscript

I have no concerns with the current state of the manuscript and I feel that the authors have addressed all my concerns

Best regards,